# All-polymer particulate slurry batteries

Wen Yan[1], Caixing Wang[1], Jiaqi Tian[1], Guoyin Zhu[1], Lianbo Ma[1], Yanrong Wang[1], Renpeng Chen[1], Yi Hu[1], Lei Wang[1], Tao Chen[1], Jing Ma[1] & Zhong Jin [1]

Redox flow batteries are promising for large-scale energy storage, but some long-standing problems such as safety issues, system cost and cycling stability must be resolved. Here we demonstrate a type of redox flow battery that is based on all-polymer particulate slurry electrolytes. Micro-sized and uniformly dispersed all-polymer particulate suspensions are utilized as redox-active materials in redox flow batteries, breaking through the solubility limit and facilitating the application of insoluble redox-active materials. Expensive ion-exchange membranes are replaced by commercial dialysis membranes, which can simultaneously realize the rapid shuttling of $H^+$ ions and cut off the migration of redox-active particulates across the separator via size exclusion. In result, the all-polymer particulate slurry redox flow batteries exhibit a highly reversible multi-electron redox process, rapid electrochemical kinetics and ultra-stable long-term cycling capability.

[1] Key Laboratory of Mesoscopic Chemistry of MOE, Jiangsu Key Laboratory of Advanced Organic Materials, School of Chemistry and Chemical Engineering, Nanjing University, Nanjing 210023, China. Correspondence and requests for materials should be addressed to Z.J. (email: zhongjin@nju.edu.cn)

With the booming development of distributed power generation, such as solar and wind power plants, large-scale energy storage technologies are required to stabilize the load-leveling of power grid system[1,2]. Redox flow batteries (RFBs) hold great promise for large-scale energy storage, owing to their unique cell architecture based on liquid electrolyte reservoirs and pumps that bring the advantages of easy scalability and excellent modularity[3–9]. In existing RFBs, all-vanadium RFBs (VFBs) are most extensively studied due to good reversibility and large power density[10–12]. However, the application of VFBs is still constrained by the high cost (~80 USD kWh$^{-1}$)[10], strong toxicity of vanadium electrolyte, and restricted operating temperature range (10–40 °C)[9]. The commercialization of VFBs is also hindered by several shortcomings of ion-exchange membrane separators, such as limited ion selectivity, high price (500–700 USD m$^{-2}$), and decreased ion conductivity at elevated operation temperature or low hydration state[11–13]. Moreover, the unrestricted diffusion of redox-active species through the separator membrane often causes reduced Coulombic efficiency and shortened cycling life. For improving the reliability of RFBs, new redox-active materials, separators and advanced system configurations are needed. Recently, increasing attention on the research of RFBs is geared towards organic- and polymer-based redox pairs[8,9,14–25]. The structural diversity and abundant sources of organic molecules enable the convenient tailoring of electrochemical properties by introducing different functional groups[16]. Because of these benefits, a variety of redox-active organic compounds have been investigated in RFBs. However, organic-based redox species often suffer from low solubility, poor chemical instability and unwanted side reactions[9]. A class of RFBs based on combining redox-active polymers or colloids with a size-exclusion separator could inhibit the crossover of active species, improve the cycling stability and reduce the battery cost[23,26].

Generally, the design of redox-active materials for RFBs should consider the following requirements[8,9]: (a) the redox conversion of active materials should be electrochemically reversible and durable within a relatively-large voltage window; (b) the homogeneous dispersion and stability of active materials in electrolyte are essential to acquire high utilization ratio, fast kinetics and long cycling life; (c) the synthesis of materials should be economical and scalable. Given the fact that a large number of redox-active materials are not soluble in water and the utilization of organic solvents may result in low ion mobility and poor rate capability[9], it is challenging to design ideal redox-active materials that satisfy all of the above demands.

To alleviate these issues, herein, we propose the design of a type of RFBs based on aqueous-dispersed all-polymer particulate slurry electrolytes with multi-electron redox capability and fast charge transfer. The all-polymer particulate slurry RFBs (APPSBs) make use of homogeneously-dispersed, micro-sized polymer particulates as redox-active species, breaking through the solubility limit of active materials and also facilitating the application of insoluble redox-active materials in RFBs. Moreover, the viscosity of electrolyte is greatly reduced in comparison with soluble organic polymers, owing to the relatively weak interactions between solvent molecules and suspended particulates. The application of particulate slurry electrolytes in RFBs makes it possible to replace expensive ion-exchange membranes with much cheaper commercial dialysis membranes through the mechanism of size exclusion[23,26]. The combination of particulate slurry electrolytes and dialysis membrane promotes the rapid shuttling of H$^+$ ions and minimizes the crossing-over of active species through the separator, contributing to the significantly improved electrochemical kinetics and cycling stability.

## Results

**Synthesis and characterizations of polymer particulates.** As a proof-of-concept demonstration, polyhydroquinone (PHQ) and polyimide (PI) are employed as the redox-active materials in particulate slurry catholyte and anolyte, respectively (Fig. 1). Different from incorporating redox molecules as pendants into inactive polymer colloids, we utilize polymers with redox-active sites on their main-chain aromatic rings. Quinone compounds, as common redox-active moieties in nature, also play an important role in the electron-transport processes of biological systems[14]. In addition to the abundant source and low toxicity, some quinone compounds exhibited outstanding performances of high electron-transfer rates and large capacities in electrochemical processes[6,17–19,25,27]. PHQ, as a redox-active polymer composed of

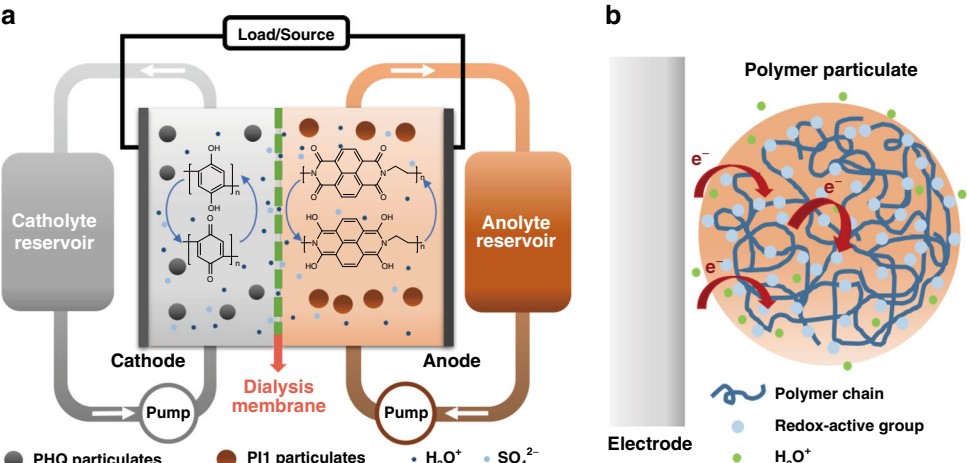

**Fig. 1** Schematic diagram of all-polymer particulate slurry batteries. **a** Schematic configuration of polyhydroquinone (PHQ)/naphthalene-1,4,5,8-tetracarboxylic acid dianhydride-ethylene diamine copolymer (PI1) all-polymer particulate slurry redox flow battery (APPSB). The PHQ/PI1 APPSB mainly consists of two electrolyte reservoirs, two peristaltic pumps and an electrochemical cell where the redox reactions take place. The particulate slurry catholyte and anolyte are separated by commercial dialysis membrane. The electrolytes are circulated between the electrochemical cell and the storage reservoirs during the charging/discharging processes. **b** Schematic diagram of proposed site-hopping mechanism to elucidate the charge transfer of particulates in the redox processes

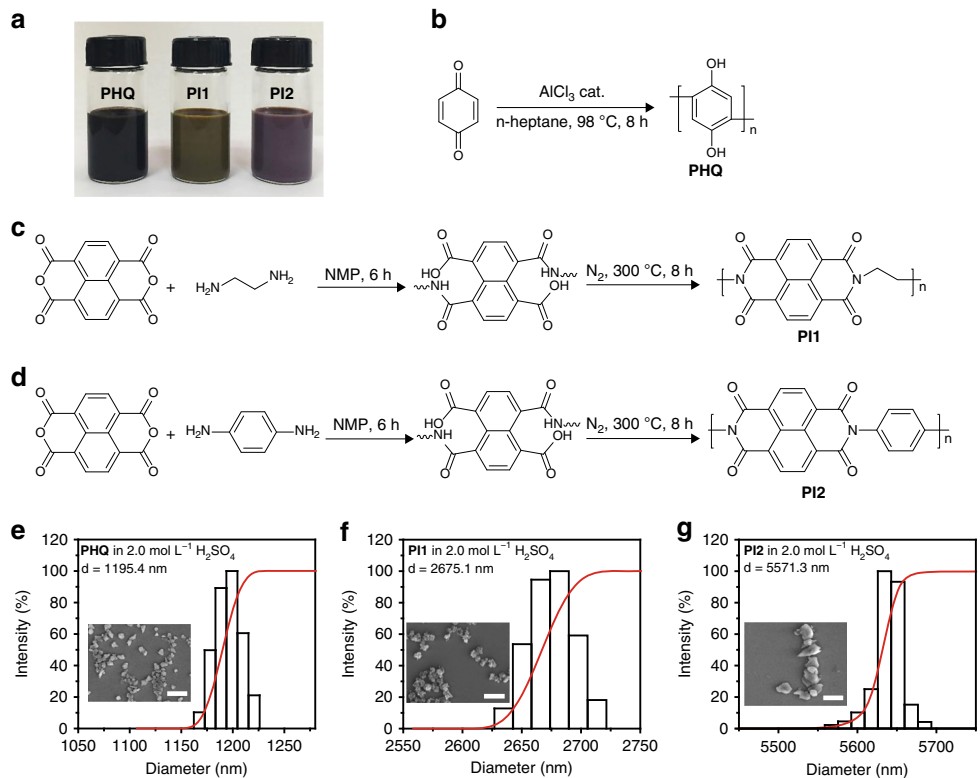

**Fig. 2** Synthesis processes and size distribution of redox-active polymer particulates. **a** Optical photographs of as-prepared 1.0 mol L$^{-1}$ (based on the mole number of repeating units) polyhydroquinone (PHQ), naphthalene-1,4,5,8-tetracarboxylic acid dianhydride-ethylene diamine copolymer (PI1) and naphthalene-1,4,5,8-tetracarboxylic acid dianhydride-p-phenylenediamine copolymer (PI2) particulate slurries. **b–d** Synthesis routes of PHQ, PI1, and PI2. **e–g** DLS diameter distributions of the polymer particulates. The inserts are SEM images of PHQ, PI1, and PI2 particulates, respectively. Scale bar, 5 μm

hydroquinone units[28], well maintains the electrochemical properties of quinone and hydroquinone, and exhibits high capacities and good stability. On the other hand, polyimide (PI) is well-known for its good thermal stability and chemical resistance originated from its rigid main chain. Although usually regarded as an insulator, PI is actually electrochemically active due to the good redox reversibility of imide groups[29,30]. Based on the differences of precursor dianhydride and diamine species, the molecular structures of PI can be broadly varied, such as PI1 (naphthalene-1,4,5,8-tetracarboxylic acid dianhydride-ethylene diamine copolymer) and PI2 (naphthalene-1,4,5,8-tetracarboxylic acid dianhydride-p-phenylenediamine copolymer). During electrochemical processes, each repeating unit of PI can exhibit a reversible multi-electron redox behavior[31,32]. In this study, PHQ and PI are introduced as active materials in RFBs. Both of these polymers can perform rapid multi-electron transfer via polymer chains in the electrochemical processes, and also have the merits of easy-to-synthesize, high chemical stability, low toxicity and, cheap cost. We find that they can be employed as a superb redox pair for APPSBs, owing to the high dispersibility, good redox reversibility and suitable electrochemical window of PHQ and PI particulates dispersed in acidic aqueous media.

As illustrated in Fig. 2 and detailed in the Methods section of Supplementary Information, PHQ and PI (including PI1 and PI2) particulates were prepared and dispersed in H$_2$SO$_4$/H$_2$O to form aqueous slurries. PHQ was synthesized via a one-step polymerization reaction of 1,4-benzoquinone[33] (Fig. 2b). PI1 was synthesized by using dianhydride and diamine as the starting materials[29] (Fig. 2c). As a control sample, PI2 was synthesized via a similar route[29] (Fig. 2d). For the sake of homogeneous dispersion and high energy density of RFBs, the polymer particulate slurries with high concentration (1.0 mol L$^{-1}$, based

on the mole number of repeating units) were prepared by acid treatment and ultrasonication (Fig. 2a). After dialysis against H$_2$SO$_4$ solution, any possibly remained monomers, oligomers or tiny polymer particles were removed, and the slurries only consisted of polymer particulates. As shown in Supplementary Fig. 1, the polymer particulates (1.0 mol L$^{-1}$, based on the mole number of repeating units) can be homogeneously dispersed in sulfuric acid (2.0 mol L$^{-1}$) under static condition at room temperature (25 °C) and low temperature (4 °C) and remain stable for at least 3 days. Moreover, the APPSBs tests were performed with the rapid circulation of electrolytes driven by peristaltic pumps, so the precipitation of polymer particulates was very minimal under the flow mode. The increase of sulfuric acid concentration can further stabilize the dispersion of polymer particulates. Technically, an even more saturated polymer particulate slurry can be prepared with concentrated sulfuric acid. However, to avoid the corrosion of the RFB system, 2.0 mol L$^{-1}$ of sulfuric acid concentration is chosen for the use in APPSBs. In contrast, for monomers, the hydroquinone monomer has a solubility of ~0.6 mol L$^{-1}$ in water at 25 °C but is susceptible to rapid oxidation in the air; imide monomers are only slightly soluble in water; and the solubility of phthalimide is less than 0.007 mol L$^{-1}$ in water at 20 °C. On the other hand, the PHQ or PI particulates dispersed in pure water without H$_2$SO$_4$ will precipitate rapidly. Hence, the micro-sized and uniformly-dispersed polymer particulate slurry electrolytes in H$_2$SO$_4$ solution can certainly break the solubility limit and facilitate the application of insoluble redox-active polymers in RFBs.

The diameter distributions of PHQ, PI1, and PI2 particulates were measured by dynamic light scattering (DLS) after diluted to 0.001 mol L$^{-1}$, as shown in Fig. 2e−g. The average sizes of polymer particulates were determined to be 1,195 nm for PHQ,

2675 nm for PI1, and 5,571 nm for PI2, respectively. Meanwhile, the kinematic viscosity of high concentration (1.0 mol L$^{-1}$) particulate slurries was measured to be 7.8 mPa s for PHQ, 3.1 mPa s for PI1 and 2.3 mPa s for PI2, respectively. Compared to high-viscosity branched polymers[23], the PHQ/PI polymer particulate slurries have reduced viscosity, thus can improve the diffusion kinetics and reduce the mass transfer resistance. The polymer particulate slurries have high concentration, good dispersity, and homogeneity. Especially, the sizes of polymer particulates are in microscale, so they can be effectively blocked by dialysis membranes with nanoscale pores via size exclusion. The featured morphologies of PHQ, PI1, and PI2 particulates were also characterized by scanning electron microscopy (SEM), showing their irregular shape, rough surface and particle sizes that match with DLS analysis (Fig. 2b−d). In Supplementary Fig. 2, the Zeta potentials of the particulate suspensions were measured to be −38.7 mV for PHQ, 48.6 mV for PI1 and 10.6 mV for PI2, respectively, indicating the dispersibility and stability of PHQ and PI1 particulate suspensions are higher than those of PI2. Fourier transform infrared spectroscopy (FTIR) analysis revealed the vibration modes of characteristic functional groups, confirming the successful preparation of these polymers (Supplementary Fig. 3). In addition, there is no discernible change in characteristic FTIR bands of these polymer particulates after acid treatment, indicating the good stability that beneficial to the cycling stability of RFBs.

**Electrochemical tests.** The electrochemical properties of PHQ and PI particulate suspensions were analyzed by cyclic voltammetry (CV) (Fig. 3a). The PHQ particulate suspension exhibited a reduction potential at 0.55 V and an oxidation potential at 0.84 V vs. standard hydrogen electrode (SHE, all of the following

electrode potentials are relative to SHE), which is in accordance with the value in literature[28,33]. The corresponding redox reaction of PHQ is displayed in Fig. 3b. Every structural unit of PHQ molecular chain can undergo a two-electron redox conversion, thus offering a theoretical specific capacity of 496 mA h g$^{-1}$. Due to the fast transformation between radical anions and dianions, there is only a broad peak rather than two distinctly-separated peaks in the oxidation or reduction process of PHQ (Fig. 3a). As shown in Supplementary Fig. 4, the CV curve of PHQ particulate suspension at the 50th cycle was nearly overlapped with that at the 1st and 2nd cycles, while the ratio of oxidation capacity to reduction capacity ($Q_1/Q_2$) were changed at different scan rates. The $Q_1/Q_2$ is 1.08 at 0.1 V s$^{-1}$ and 1.34 at 0.025 V s$^{-1}$, respectively. The small oxidation peak at 1.2 V (vs. SHE) is proposed to be originated from the electro-polymerization of PHQ[33,34]. As illustrated in Supplementary Fig. 5, when the terminal hydroquinone was oxidized to the protonated benzoquinone in the electro-oxidation process, it might react with the non-protonated hydroquinone, leading to the electro-polymerization of polymer chains. The electro-polymerization can provide additional oxidation capacity, but it doesn't affect the reversibility of redox-active groups. The parasitic reaction of electro-polymerization can be suppressed by increasing the current rate, as indicated by the Tafel plots (Supplementary Fig. 6). To investigate the charge transfer mechanism of particulates, the electrochemical properties of benzoquinone monomer (BQ) and PHQ particulates with different sizes were compared. PHQ with average diameters of 1 μm and 50 μm are termed as PHQ-1 and PHQ-50 (Supplementary Fig. 7). As shown in Supplementary Fig. 8a, with the decrease of particulate size, the electrochemical polarization (ΔE) of PHQ particulates was reduced from 618 mV (PHQ-50) to 200 mV (PHQ-1), indicating the sluggish charge transfer within and among PHQ particulates was improved. At the scan rate of

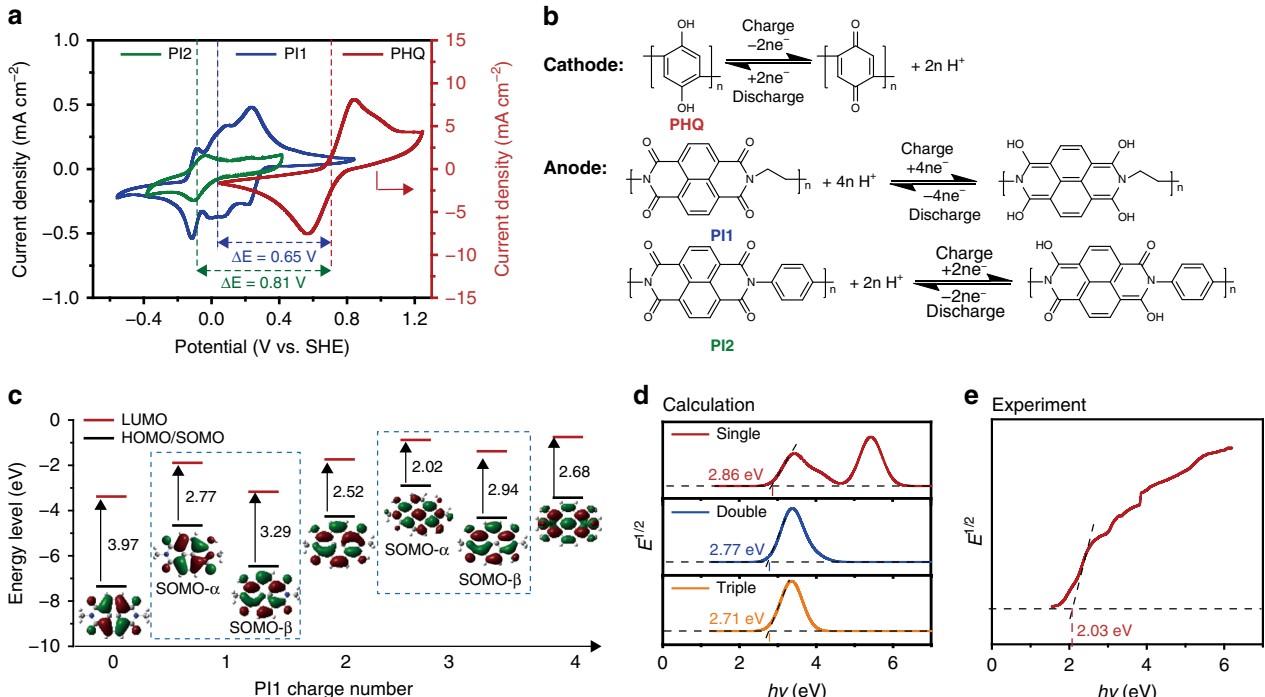

**Fig. 3** Electrochemical redox properties of polymer particulate suspensions. **a** CV curves of polyhydroquinone (PHQ), naphthalene-1,4,5,8-tetracarboxylic acid dianhydride-ethylene diamine copolymer (PI1) and naphthalene-1,4,5,8-tetracarboxylic acid dianhydride-p-phenylenediamine copolymer (PI2) particulate suspensions (containing 0.005 mol L$^{-1}$ redox-active polymer and 2.0 mol L$^{-1}$ H$_2$SO$_4$) at a scan rate of 0.025 V s$^{-1}$ for all the cases. **b** Reversible multi-electron redox conversion of PHQ, PI1, and PI2. **c** Theoretical highest occupied molecular orbital (HOMO)/singly occupied molecular orbital (SOMO), lowest unoccupied molecular orbital (LUMO) energy levels as well as band gaps of PI1. **d** DFT-calculated Tauc plots of single-, double-, and triple-unit PI1 molecules and their optical band gaps. **e** Experimental Tauc plot of PI1 polymer particulates

0.025 V s$^{-1}$, the CV oxidation peak currents of BQ, PHQ-1 and PHQ-50 are 0.74, 0.38, and 0.25 mA cm$^{-2}$, respectively, indicating the large particle size has negative effect to the utilization ratio of redox-active species[35]. When the scan rate is decreased to 0.006 V s$^{-1}$, the CV oxidation peak currents of BQ, PHQ-1 and PHQ-50 are 0.10, 0.08, 0.06 mA cm$^{-2}$, respectively (Supplementary Fig. 8b). These results indicate that the smaller particle size may lead to more charge transfer inside the particulates and more units involved in the redox reaction. Based on the theoretical model proposed in the literatures[26,35,36], a site-hopping mechanism is proposed to elucidate the charge transfer of particulates during redox processes. As shown in Fig. 1b, the redox-active sites on particulate surface are firstly reduced when approaching to the electrode. The electrons can transport through the interface by collision between the polymer particulates and current collectors with the assistance of electric double layer (EDL) formed on the surface of particulates. Then, the charges transport across the polymer chains by electron hopping between the highly populated redox-active groups[26,36,37]. Meanwhile, the H$^+$ ions in the electrolyte can permeate into the polymer particulates, accelerating the charge transfer in the redox processes[38]. The CV curve of PI1 shows four peaks at 0.23, 0.07, 0.00, −0.12 V in the reduction process, and four peaks at 0.28, 0.12, 0.05, −0.06 V in the oxidation process, respectively (Fig. 3a). The redox peaks of PI1 can be ascribed to the enolization and its inverse process of carbonyl groups[29]. The CV signal splits of PI1 might be originated from the formation of resistive diffusion layers of intermediates near the electrode surface. Each repeating unit of PI1 can carry out a reversible four-electron redox process, offering a specific capacity of 400 mA h g$^{-1}$ theoretically.

To examine the multi-electron redox mechanism of PI1 during electrochemical process, the frontier molecular orbitals of PI1 were studied by density functional theory (DFT) calculations. Firstly, the computed vibrational frequencies of PI1 are in good agreement with the experimental FTIR spectrum (Supplementary

Fig. 3b and Supplementary Fig. 9), thus allowing the application of B3LYP/6-31+G (d, p) basis set. The locations of valence-shell electrons in the molecule were also studied to provide qualitative information on its electronic structure[39]. In Fig. 3c, the plots of highest occupied molecular orbitals (HOMO) retain within the PI1 units during the redox process, indicating the good stability of PI1 and its redox intermediates. Moreover, the energy levels of HOMO and LUMO (lowest occupied molecular orbitals), as well as the HOMO-LUMO band gaps, were computed. The HOMO-LUMO band gaps of PI1 units during the redox process were calculated to be in the range of 2–4 eV from the HOMO-LUMO plots (Fig. 3c). On the other hand, the optical band gaps of PI1 determined by the calculated UV-Vis spectra were 2.86 eV for the single-unit molecule, 2.77 eV for the double-unit molecule and 2.71 eV for the triple-unit molecule (Fig. 3d). The experimental optical band gap of the PI1 polymer determined by UV-Vis absorption spectroscopy is 2.03 eV (Fig. 3e). Both theoretical and experimental results indicate that the optical band gaps of PI1 become narrower along with the increase of unit number.

According to the Randles-Sevcik equation, a linear increase of the peak current (i) against the square root of the scan rate ($v^{1/2}$) is observed for PHQ (Fig. 4a and Supplementary Fig. 10a) and PI1 (Fig. 4d and Supplementary Fig. 10d), respectively, indicating the occurrence of charge diffusion inside the particulates rather than only on the surface. Rotating-disk-electrode (RDE) voltammetry was performed to investigate the kinetic properties of PHQ and PI1 (Fig. 4b, e) particulate suspensions. The Levich analysis of RDE voltammograms, obtained from a variety of rotation speeds, yielded the diffusion coefficients (D) of $9.20 \times 10^{-7}$ cm$^2$ s$^{-1}$ for PHQ and $1.66 \times 10^{-7}$ cm$^2$ s$^{-1}$ for PI1 (Supplementary Fig. 10b, e). Subsequent Koutecký–Levich analysis revealed the mass-transport-independent currents, which were fitted to the Butler-Volmer equation to obtain the electron-transfer rate constants ($k_0$). The Tafel slope was 2.07 V$^{-1}$ for PHQ and

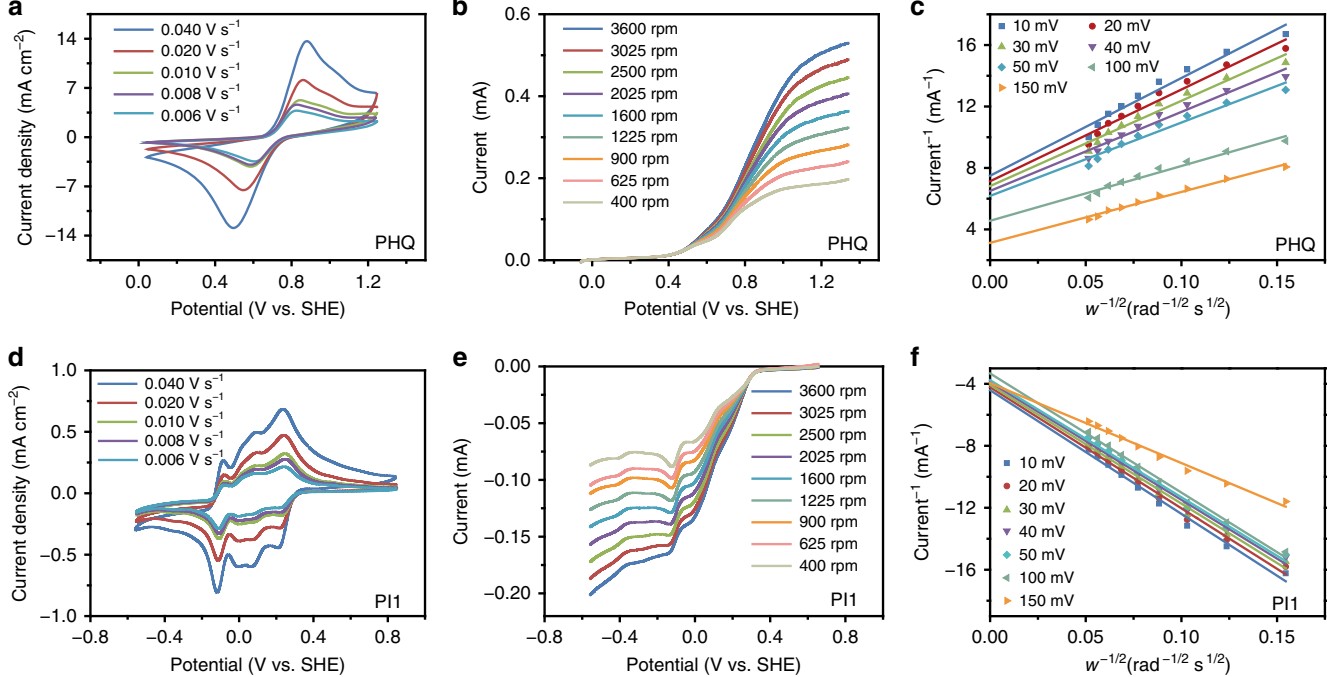

**Fig. 4** Electrochemical characterizations of polymer particulate suspensions. **a**, **d** Cyclic voltammograms of 0.1 mol L$^{-1}$ polyhydroquinone (PHQ) or naphthalene-1,4,5,8-tetracarboxylic acid dianhydride-ethylene diamine copolymer (PI1) particulates in 2.0 mol L$^{-1}$ H$_2$SO$_4$ aqueous solution at different scan rates. **b**, **e** RDE measurements of 0.005 mol L$^{-1}$ PHQ or PI1 particulates in 2.0 mol L$^{-1}$ H$_2$SO$_4$ aqueous solution at rotating electrode speeds from 400 rpm to 3600 rpm. **c**, **f** Koutecký–Levich plot of PHQ or PI1 particulates

1.73 $V^{-1}$ for PI1; and the electron-transfer coefficient ($\alpha$) was calculated as 0.878 for PHQ and 0.900 for PI1. The $k_0$ was calculated to be $6.72 \times 10^{-4}$ cm $s^{-1}$ for PHQ (Fig. 4c and Supplementary Fig. 10c) and $2.31 \times 10^{-3}$ cm $s^{-1}$ for PI1 (Fig. 4f and Supplementary Fig. 10f), respectively. The $D$ and $k_0$ of PHQ and PI1 particulate suspensions are comparable to $VO^{2+}/VO^{+}$, $V^{3+}/V^{2+}$ and other previously-reported redox-active organic molecules summarized in Supplementary Table 1[17,20–24,26,40–43]. The high diffusion and electron-transfer rates of PHQ and PI particulate suspensions lead to improved electrochemical kinetics, which is conducive to the rate performance for high-power output applications.

**Battery performances.** To investigate the capability of PHQ and PI1 as an electrochemical redox couple, constant-current charge/discharge tests were conducted within a flow battery system. As shown schematically in Fig. 1, the catholyte and anolyte are 1.0 mol $L^{-1}$ PHQ and PI1 particulate slurries in 2.0 mol $L^{-1}$ sulfuric acid solution, respectively. The cathode and anode compartments are separated by a dialysis membrane with a molecular weight cutoff of 1000 g $mol^{-1}$ (MWCO1,000). As an alternative to ion-exchange membranes, dialysis membrane was used as separator, which could simultaneously realize the rapid shuttling of $H^{+}$ ions and the effective blocking of polymer particulates. The detailed configurations of the RFB system are presented in Supplementary Fig. 11. As shown in the Nyquist plot (Supplementary Fig. 12), the cell resistance of PHQ/PI1 APPSBs is 2.9 Ohm, which is primarily composed of the Ohmic resistances of particulate slurries, separator, and cell stack. The high-frequency semicircle is ascribed to the charge transfer during the redox process, indicates that the charge transfer resistance is 10.7 Ohm. The low-frequency sloping line is ascribed to the ion diffusion in the electrolyte and within the particulates.

Representative charge/discharge curves of PHQ/PI1 APPSBs from 0.001 V to 1.2 V are displayed in Fig. 5a. A discharge capacity of 8.95 Ah $L^{-1}$ (74.4 Ah $kg^{-1}$ based on the weights of PHQ and PI1) was obtained at the current density of 5 mA $cm^{-2}$. For polymers with redox-active groups along the main chain, the multi-electron transfer through the molecule chain may be facilitated over a wide range of voltage. According to the multi-electron redox processes revealed by CV analysis (Fig. 3), presumably, there might be several plateaus in the charge/discharge curves. However, there is no flat plateaus but rather sideling curves in Fig. 5a, which is ascribed to the complex electron-transfer process on the surface and inside of the polymer particulates[35]. A gentle slope between 0 and 0.1 V is observed in Fig. 5a, which is based on the nearly overlapping part of the reduction of PHQ and the oxidation of PI1 in CV analysis (Fig. 3a). The average charge voltage is 0.90 V, and the average discharge voltage is 0.53 V. To increase the voltage efficiencies of PHQ/PI1 APPSBs, this part of capacities was abandoned by charging/discharging in the range from 0.1 to 1.2 V, and the voltage efficiencies increased from 58.9% to 70.0%. The capacity retention and Coulombic efficiencies of PHQ/PI1 APPSBs are presented in Fig. 5b, showing a capacity decay of 0.36% per cycle and Coulombic efficiencies of ~87%.

DLS and Zeta potential measurements reveal the similar dispersibility and stability of the diluted polymer particulate suspensions after cycling test (Supplementary Fig. 13). As shown in the SEM images (Supplementary Fig. 14a, b), most of the PHQ and PI1 particulates keep approximately the original dimensions, but some particulates aggregate into larger particles. For PI2, the aggregation is worse than PI1 (Supplementary Fig. 14c). The photographs of disassembled PHQ/PI2 APPSB cell after long-terming cycling is shown in Supplementary Fig. 14d, which shows

slight accumulation of polymer particulates on the carbon paper and in the flow channel. The agglomerates precipitated in the flow channels and reservoirs may result in capacity fading after long-term tests. Therefore, to further improve the cycling stability of APPSBs, we suggest to introduce appropriate dispersion stabilizer in the particulate suspensions without the compromise of electrochemical performances, which will be an important aspect of our future research. UV-Vis absorption spectra of the diluted polymer particulate suspensions were collected at fully charged and fully-discharged states to identify the molecular structure changes during the redox processes. As shown in Supplementary Fig. 15, the broad absorption peaks of polymer particulates are red shifted along with the increase of unit number and extend to the visible light region[30,44], which is consistent with the calculation results in Fig. 3d. The broad absorption peaks at around 300 and 320 nm of the reduced and oxidized forms of PHQ particulates are ascribed to the $\pi-\pi^{\star}$ transition of the benzene ring. After the electro-oxidation of PHQ, a new absorption peak at 246 nm was emerged (Supplementary Fig. 15a), and the absorption peak at 326 nm was enhanced and red-shifted, indicating the conjugation effect between the carbonyl groups and the backbone of benzene rings[35]. On the other hand, as shown in Supplementary Fig. 15b, the electro-reduction of PI1 leads to new absorption peaks at 304 nm, 539 nm and 645 nm[30], owing to the spatial charge distribution variation of $\pi-$conjugation system.

The rate performance of PHQ/PI1 APPSBs and representative charge/discharge curves from 5 to 20 mA $cm^{-2}$ are shown in Fig. 5c, d, exhibiting discharge capacities of 7.41 Ah $L^{-1}$ (4.45 Wh $L^{-1}$, 5 mA $cm^{-2}$), 5.67 Ah $L^{-1}$ (3.39 Wh $L^{-1}$, 10 mA $cm^{-2}$), 4.95 Ah $L^{-1}$ (2.87 Wh $L^{-1}$, 15 mA $cm^{-2}$), 3.95 Ah $L^{-1}$ (1.98 Wh $L^{-1}$, 20 mA $cm^{-2}$). The Coulombic efficiencies are 85.6% (5 mA $cm^{-2}$), 91.5% (10 mA $cm^{-2}$), 95.6% (15 mA $cm^{-2}$), and 99.1% (20 mA $cm^{-2}$). The increasing Coulombic efficiency with increasing current densities may be attributed by the aforementioned suppression of the parasitic electro-polymerization reaction of PHQ at high current densities (Supplementary Figs. 5, 6). The voltage efficiencies are 70.0% (5 mA $cm^{-2}$), 66.4% (10 mA $cm^{-2}$), 62.3% (15 mA $cm^{-2}$), 58.6% (20 mA $cm^{-2}$), and the energy efficiencies are 59.9% (5 mA $cm^{-2}$), 60.8% (10 mA $cm^{-2}$), 59.5% (15 mA $cm^{-2}$), 58.1% (20 mA $cm^{-2}$). As shown in Fig. 5e, the long-term cycling test of PHQ/PI1 APPSBs with dialysis membrane separator at 20 mA $cm^{-2}$ shows an initial discharge capacity of 4.95 Ah $L^{-1}$ with Coulombic efficiencies close to 100% and energy efficiencies around 60%. The discharge capacity retention after 300 cycles is 70% of the initial capacity, corresponding to a low capacity decay of 0.1% per cycle.

Compared to the PHQ/PI1 APPSBs with dialysis membrane separators, those with Nafion-117 membrane separators exhibited inferior performances. Firstly, electrochemical impendence spectroscopy (EIS) measurements were performed to compare the ionic conductivity of MWCO1,000 dialysis membrane and Nafion-117 membrane. After soaking in 2.0 mol $L^{-1}$ $H_2SO_4/H_2O$ solution electrolyte, the electrical resistance was determined to be 0.900 $\Omega$ for MWCO1,000 dialysis membrane and 1.289 $\Omega$ for Nafion-117 membrane (Supplementary Fig. 16). According to the equation of ion conductivity (shown in the Methods Section of Supplementary Information), the ion conductivity of MWCO1,000 dialysis membrane was calculated as 0.054 S $cm^{-1}$, which is higher than that of Nafion-117 membrane (0.011 S $cm^{-1}$).

To compare the electrochemical performances of PHQ/PI1 APPSBs with dialysis membrane or Nafion-117 membrane separators, long-term cycling tests were conducted with 0.1 mol $L^{-1}$ polymer particulate suspensions at 20 mA $cm^{-2}$ under static operation. As shown in Supplementary Fig. 17, the PHQ/PI1 APPSBs with dialysis membrane separator shows an initial capacity of 0.81 Ah $L^{-1}$ and exhibits a stable charge/discharge capability for 5000 cycles with energy efficiencies higher than

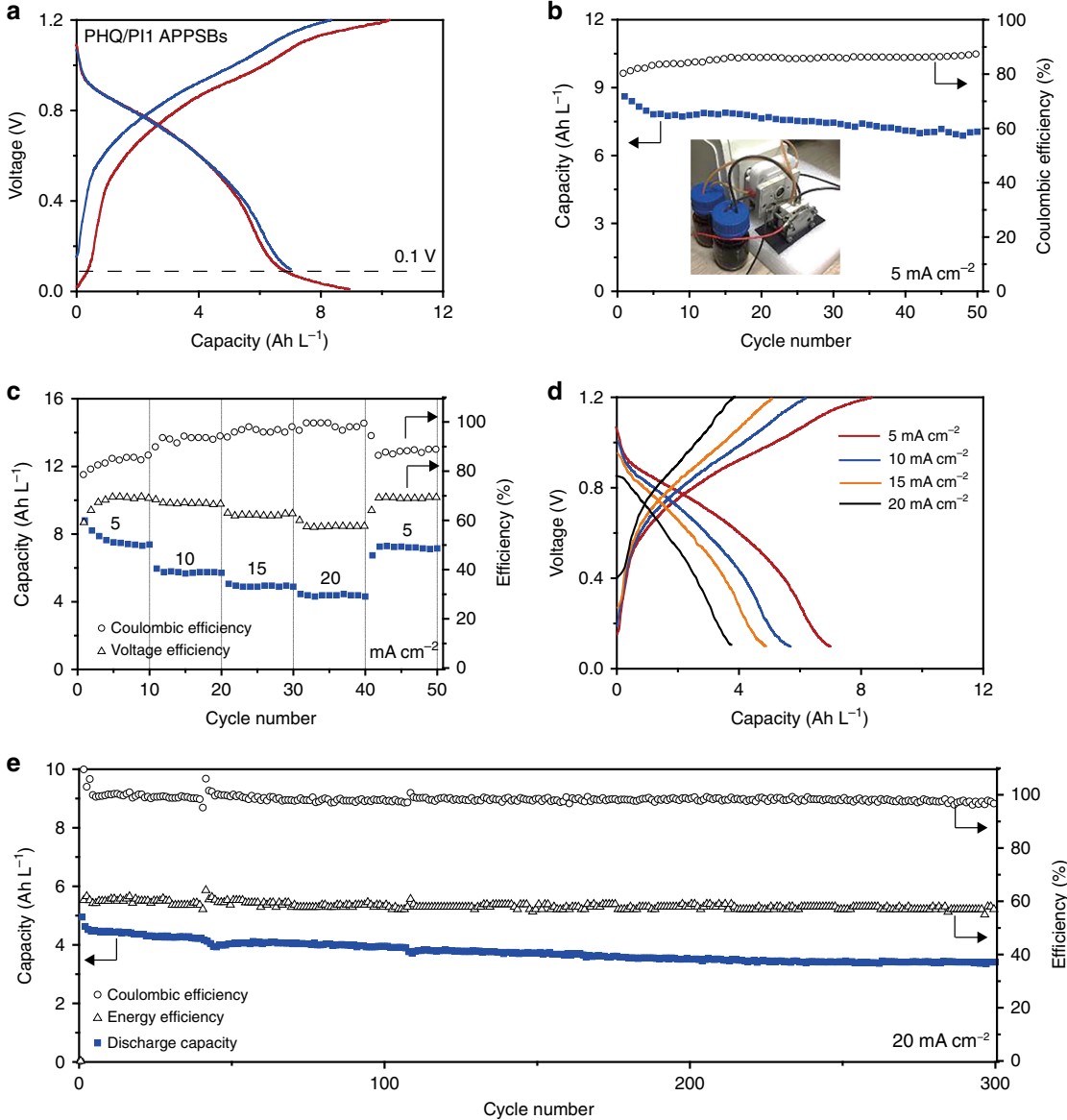

**Fig. 5** Electrochemical performances of all-polymer particulate slurry batteries. **a** Representative charge/discharge curves of polyhydroquinone (PHQ), naphthalene-1,4,5,8-tetracarboxylic acid dianhydride-ethylene diamine copolymer (PI1) all-polymer particulate slurry redox flow battery (APPSB) with dialysis membrane separators at 5 mA cm$^{-2}$. The catholyte and anolyte are 1.0 mol L$^{-1}$ PHQ and PI1 particulate slurries in 2.0 mol L$^{-1}$ sulfuric acid solution, respectively. **b** Stability test of PHQ/PI1 APPSBs at 5 mA cm$^{-2}$. The insert is an optical photograph of the APPSBs system. **c** Rate performance, Coulombic efficiencies and voltage efficiencies from 5 mA cm$^{-2}$ to 20 mA cm$^{-2}$. **d** Representative charge/discharge curves of PHQ/PI1 APPSBs at different current densities. **e** The long-term stability test of PHQ/PI1 APPSBs studied by repeating charge/discharge cycles at 20 mA cm$^{-2}$

50%. The discharge capacity retention after 5000 cycles is 74% of the initial capacity, corresponding to a low capacity decay of only 0.0052% per cycle and 1.83% per day. In contrast, the PHQ/PI1 APPSBs with Nafion-117 membrane separator shows a slightly-lower initial capacity (0.77 Ah L$^{-1}$), and the discharge capacity declines to 42% of its initial value after 1400 cycles (Supplementary Fig. 17), indicating inferior stability compared to that with dialysis membrane. Initially, the energy efficiency of PHQ/PI1 APPSBs with Nafion-117 membrane (51.1%) is comparable to that with dialysis membrane (51.6%), but it declines to ~30% after 1400 cycles. Notably, the Coulombic efficiencies of PHQ/PI1 APPSBs with dialysis membrane and Nafion-117 separators both remain above 95% throughout long-term cycling (Supplementary Fig. 17), indicating the remarkable reversibility of PHQ/PI1 redox couple. It can be concluded that the decreases of capacity and energy efficiency of PHQ/PI1 APPSBs with Nafion-117

membrane are originated from the impedance growth and performance degradation of Nafion membrane, rather than true capacity fade. The ultralong cycling life of PHQ/PI1 APPSBs with dialysis membrane separator indicates the excellent redox stability of polymer particulate slurries, and also verifies the outstanding durability of dialysis membrane separator. Benefited from the superior ion-conducting property, the dialysis membrane exhibited lower resistance compared to Nafion-117 membrane, which is similar to nanofiltration membrane[13]. The relatively thick Nafion membrane leads to lower ion permeability and higher area resistance, which also limits the voltage efficiency of RFBs[45]. In result, the replacement of Nafion membrane with dialysis membrane in particulate APPSBs can greatly improve the overall performances, and also reduces the system cost of battery since the price of dialysis membranes is only about one-tenth of Nafion membranes.

PI2, as another kind of PI, was also compared as redox-active anode material in APPSBs. The CV test of PI2 particulate suspension revealed a reversible redox reaction at around −0.10 V in the anodic region (Fig. 3a). The corresponding redox reaction of PI2 are shown in Fig. 3b. In theory, the replacement of alkyl group (PI1) with phenyl group (PI2) can lower the energy level of lowest unoccupied molecular orbital (LUMO) and makes the polymer more reductive, thus the operation voltage could be boosted[29,39]. However, owing to the limit of hydrogen evolution potential in acidic aqueous systems, the electrochemical redox reaction of PI2 within this voltage window was estimated to be an incomplete two-electron process, rather than a complete four-electron process like PI1. As shown in Fig. 3a, the PHQ/PI2 APPSBs provide an open-circuit voltage of 0.81 V, which is higher than that of PHQ/PI1 APPSBs (0.65 V). This suggests the functional group modification of redox-active polymers is a promising approach for further tuning their electrochemical properties. From the Levich analysis of RDE voltammograms (Supplementary Fig. 18), the diffusion coefficient ($D$), Tafel slope, electron-transfer coefficient ($\alpha$) and electron-transfer rate constant ($k_0$) of PI2 particulate suspension was determined to be $7.01 \times 10^{-8}$ cm$^2$ s$^{-1}$, 7.04 V$^{-1}$, 0.591 and $1.01 \times 10^{-4}$ cm s$^{-1}$, respectively. Significant difference of redox kinetics of PI1 and PI2 particulates may be partially originated from their different pore structures. Brunauer−Emmer−Teller (BET) analysis shows the specific surface areas of 19.9 and 2.0 m$^2$ g$^{-1}$ for PI1 and PI2 particulates, respectively (Supplementary Fig. 19). The pore size distribution demonstrates the presence of some mesopores of PI1, although the pore volume is relatively low. Compared to the compact stacking structure of PI2 resulted from the rigid benzene rings, the loosely-stacking structure of PI1 attributed by the flexible alkyl segments on the main chains may accelerate the transfer of abundant electrolyte ions into the particulates.

The charge/discharge capability of PHQ/PI2 APPSBs was also investigated, as detailed in Supplementary Fig. 20, exhibiting an average discharge voltage (0.70 V) higher than that of PHQ/PI1 APPSBs (0.53 V), an initial capacity of 5.38 Ah L$^{-1}$ (3.69 Wh L$^{-1}$) at 5 mA cm$^{-2}$ and an average capacity decay of 1.06% per cycle. When charging/discharging in the range from 0.1 to 1.2 V, the voltage efficiency increased from 76.1% to 78.3%. The rate performance and representative charge/discharge curves of PHQ/PI2 APPSBs from 5 mA cm$^{-2}$ to 20 mA cm$^{-2}$ are shown in Supplementary Fig. 20c, d. The discharging capacities are 4.30 Ah L$^{-1}$ (3.02 Wh L$^{-1}$, 5 mA cm$^{-2}$), 3.5 Ah L$^{-1}$ (2.32 Wh L$^{-1}$, 10 mA cm$^{-2}$), 2.81 Ah L$^{-1}$ (1.83 Wh L$^{-1}$, 15 mA cm$^{-2}$), 2.32 Ah L$^{-1}$ (1.48 Wh L$^{-1}$, 20 mA cm$^{-2}$), respectively. The Coulombic efficiencies are 84.2% 5 mA cm$^{-2}$), 90.8% (10 mA cm$^{-2}$), 93.7% (15 mA cm$^{-2}$), 95.6% (20 mA cm$^{-2}$). The voltage efficiencies are 78.3% (5 mA cm$^{-2}$), 75.1% (10 mA cm$^{-2}$), 72.6% (15 mA cm$^{-2}$), 65.0% (20 mA cm$^{-2}$) and the energy efficiencies are 65.9% (5 mA cm$^{-2}$), 68.2% (10 mA cm$^{-2}$), 68.0% (15 mA cm$^{-2}$), 62.1% (20 mA cm$^{-2}$). In this work, the energy efficiencies of APPSBs are round 60–70%, which is comparable to other polymer-based RFBs reported in literatures[23,46–48]. Compared with PI1, the higher discharge voltage of PI2 conforms that the reduction potential of PI2 is more negative than PI1, resulting in smaller overlapping area with PHQ in CV analysis. However, the relatively inferior dispersibility and stability of PI2 particulate slurries have a negative effect on the utilization ratio of active materials and the long-term cycling performance of PHQ/PI2 APPSBs. The cycling performance of PHQ/PI2 APPSBs with dialysis membrane or Nafion-117 membrane separator is compared in Supplementary Fig. 21. For PHQ/PI2 APPSBs with dialysis membrane separator, a total retention of 75% of discharge capacity was preserved after 1000 charge/discharge cycles. Once again, the long-term cycling stability of PHQ/PI2 APPSBs with dialysis membrane separator is found to be superior to those with Nafion-117 membrane separator. Through the control experiments of PI2, we conclude that the electrochemical properties of redox-active polymers could be finely tailored to a large extent by modifying the molecular structures. The influences of particle size on the electrochemical and physicochemical properties of polymer particulates were investigated. Briefly, ballmilling processes were performed to further decrease the size of PI1 and PI2 particulates, and the control samples after ballmilling for 48 h were termed as PI1-ballmilled, PI2-ballmilled, respectively. As shown in the SEM images and DLS curves (Supplementary Figs. 22, 23), the average particle size of PI1 decreased from 2.7 μm (PI1) to 0.8 μm (PI1-ballmilled), and the average particle size of PI2 decreased from 5.6 μm (PI2) to 0.9 μm (PI2-ballmilled). The Zeta potentials of PI1-ballmilled and PI2-ballmilled were measured to be 47.8 mV and 34.3 mV, indicating the dispersibility and stability were improved after ballmilling. CV analysis revealed the increased Faradaic response of the smaller PI1 and PI2 (Supplementary Fig. 24). Diffusion coefficient, including the physical transport of particulates to the electrode and the charge transport of the particulates, are calculated to be $1.7 \times 10^{-7}$ cm$^2$ s$^{-1}$ (PI1), $3.4 \times 10^{-7}$ cm$^2$ s$^{-1}$ (PI1-ballmilled), $0.7 \times 10^{-7}$ cm$^2$ s$^{-1}$ (PI2), and $1.3 \times 10^{-7}$ cm$^2$ s$^{-1}$ (PI2-ballmilled) (Supplementary Fig. 10, 18, 25, 26). The smaller particle size accelerates the particulate diffusion and charge transport in the redox process (Supplementary Table 2). Constant-current charge/discharge tests of PHQ/PI2-ballmilled APPSB shows discharging capacity of 8.40 Ah L$^{-1}$ (6.05 Wh L$^{-1}$) at the current density of 5 mA cm$^{-2}$, larger than that of PHQ/PI2 APPSB (4.30 Ah L$^{-1}$, 3.02 Wh L$^{-1}$), demonstrating higher capacity utilization of smaller polymer particulates (Supplementary Fig. 27). We suggest that the utilization ratio of active materials could be further improved by tuning the size, microstructure, and compositions of electrochemical-active particulates, such as constructing conductive agent composites, as well as adding proper electrolyte stabilizer without the compromise of electrochemical performances. Promoted by the good redox kinetics and reversibility of PHQ and PI, the PHQ/PI APPSBs can be stably operated for long-term cycling, delivering capacity of 4.95 Ah L$^{-1}$ (3.1 Wh L$^{-1}$) at the current density of 20 mA cm$^{-2}$ with a capacity retention of 70% after 300 cycles, which is comparable to other current state-of-art RFBs[17,20–24,26,41–43,49,50] (Supplementary Table 1).

## Discussion

In summary, we developed a type of RFB by utilizing all-polymer particulate slurries as aqueous catholyte and anolyte. The redox-active polymer particulates (PHQ and PI) prepared by simple approaches can be used as inexpensive active materials, exhibit homogeneous dispersibility, good redox reversibility, and high electrochemical stability. Furthermore, the incorporation of particulate slurry electrolytes and dialysis membrane separator effectively minimizes the crossing-over of active species via size exclusion, bringing forth much superior cycling performance than those with ion-exchange membranes. Based on these merits, the PHQ/PI APPSBs shows high Coulombic efficiency and exceptional cycling stability. Further investigation on the improvement of energy density in APPSBs will be an important aspect of our future research. Especially, in-depth studies on the rheological properties of concentrated particulate suspensions will be investigated, and the system efficiency of large-scale batteries shall be optimized to reach the best balance. We expect this work may provide new insights and inspirations for exploiting novel redox-active species and other key components in advanced energy storage systems.

## Methods

**Materials and chemicals**. All reagents were purchased from commercial sources and used as received unless stated otherwise. The carbon papers were purchased from SGL, Germany. The dialysis membranes (MWCO1,000) were purchased

from Viskase, USA. The Nafion-117 membranes were purchased from DuPont, USA.

**Synthesis of polyhydroquinone**. Equimolar p-benzoquinone (5.40 g, 0.05 mol) and anhydrous aluminum chloride (6.67 g, 0.05 mol) were reacted under reflux in 120 mL n-heptane at 98 °C for 8 h[33,51]. The mixture was filtrated, washed with water for several times to remove aluminum chloride, dried at 60 °C in air for 10 h. The final product PHQ was black powder with a yield of 70% and a number-average molecular weight ($M_n$) of ~3,400.

**Synthesis of redox-active polymer particulates**. There are two steps in the polycondensation process for the synthesis of polyimides[29]. Firstly, equimolar 1,4,5,8-naphthalenetetracarboxylic dianhydride (NTCDA, 5.36 g, 0.02 mol) and diamine (ethylene diamine (EDA, 1.20 g, 0.02 mol) for PI1, or p-phenylenediamine (pPDA, 2.16 g, 0.02 mol) for PI2 were added under reflux in 100 mL N-methylpyrrolidone (NMP) solvent at room temperature for 6 h; the mixture was filtrated, washed with ethanol for several times, dried at 120 °C in air for 12 h. The product of this solution polymerization step was polyamic acid. Secondly, to achieve a complete imidization and remove the residual solvent in the polymer, the as-obtained powder was heated in $N_2$ for 8 h at 300 °C.

**Acid treatment and dialysis**. To prepare the particulate slurries, 1.08 g of PHQ, 2.92 g of PI1 or 3.40 g of PI2 was dispersed in 4.0 mL of 5.0 mol $L^{-1}$ $H_2SO_4$, respectively. Mud-like suspensions with high concentration polymers were prepared under sonication for 1 h. After ultrasonic treatment, the above polymer muds were diluted into particulate slurries containing 1.0 mol $L^{-1}$ polymer (based on the mole number of repeating units) and 2.0 mol $L^{-1}$ $H_2SO_4$. The slurries were dialyzed with dialysis membranes (MWCO1,000) against 2.0 mol $L^{-1}$ $H_2SO_4$ solution to remove any possible monomers, oligomers, and tiny polymer particles. No obvious precipitation was observed in the particulate slurries after standing still for 72 h.

**Material characterization**. The kinematic viscosities of the suspension electrolytes were measured by a Ubbelohde viscometer at 25 °C. The size distribution of polymer particulates was determined by a dynamic laser scattering (DLS, Bruker BI-200SM) with a red laser source (640 nm) at 25 °C. The Zeta potentials of 0.001 mol $L^{-1}$ polymer particulate suspensions in 0.002 mol $L^{-1}$ $H_2SO_4$ were collected by a Malvern Zetasizer Nano Z instrument. Fourier transform infrared spectroscopy (FTIR, Bruker Vertex 70) analysis was conducted by using powdered samples (vacuum dried at 50 °C for 12 h before testing). UV-Vis absorption spectra in solid state were collected in the wavelength range of 200–800 nm using $BaSO_4$ as the matrix powder (Shimadzu UV-2600). The optical band gap of PI1 polymer was calculated according to the Tauc plot. UV-Vis absorption spectra of the diluted polymer particulate suspensions at fully-charged and fully-discharged states were measured in the wavelength range of 200–800 nm. Field-emission scanning electronic microscopy (FESEM, FEI NanoSEM-450) was performed to analyze the morphology of polymer particulates by dispersing on a silicon wafer. Nitrogen adsorption/desorption isotherms were recorded at 77 K using a Micrometrics ASAP 2020 analyzer. The surface area and pore size distribution were determined by Brunauer–Emmett–Teller (BET) and Barrett–Joyner–Halenda method, respectively.

**Electrochemical characterization**. Cyclic voltammetry (CV) tests and Rotating disk electrode (RDE) voltammetry tests were performed on an electrochemical workstation (Chenhua, CHI-760E). A standard three-electrode setup was built by using a glassy-carbon electrode (with 3 mm diameter for CV and 5 mm diameter for RDE, respectively) as working electrode, a saturated calomel electrode (SCE, 0.242 V vs. SHE) as reference electrode and a platinum wire electrode as counter electrode. For RDE tests, the rotation speed was controlled by a rotating ring-disk electrode system (ALS RRDE-3A), from 400 rpm to 3,600 rpm.

The diffusion coefficients (D) were calculated according to the Randles-Sevcik equation[23]:

$$i_p = 268,600 \times n^{2/3} AD^{1/3} cv^{1/2} \tag{1}$$

Where $i_p$ is the peak current, $n$ is the number of electrons transferred per redox reaction, $A$ is the electrode area, $v$ is the scan rate, and $c$ is the bulk concentration of the redox-active repeating structural unit of the polymer.

Through the analysis of RDE voltammograms via the Levich plot (limiting current versus $w^{1/2}$), the corresponding $D$ can be obtained from the Levich equation:

$$limiting\ current = 0.62 \times nFAD^{2/3} w^{1/2} v^{-1/6} c \tag{2}$$

Where $F = 96,485$ C $mol^{-1}$ is the Faraday's constant, $w$ is the rotation speed, and $v$ is the kinematic viscosity of the suspension. The mass-transfer-independent kinetic current $i_k$ was yielded by applying the Koutecký–Levich equation, and then

subsequently fitted by the Butler–Volmer equation:

$$1/i_k(\eta) = exp(\eta F/(RT))/(2FAk_0 c) \tag{3}$$

A Tafel plot ($log|i_k|$ versus overpotential $\eta$) fitting equation allows the determination of $i_0$ and electron-transfer rate constant $k_0$, according to the following Eqs (4) and (5), respectively:

$$log|i_k(0)| = log|i_0| \tag{4}$$

$$k_0 = i_0 / AFc_0 \tag{5}$$

The electron-transfer coefficients ($\alpha$) were determined via the slope of the Tafel plot (s):

$$s = (1 - \alpha) F / 2.3RT \tag{6}$$

Where $R$ is the universal gas constant and $T$ is the absolute temperature.

The electrochemical impendence spectroscopy (EIS) measurement of APPSBs was performed before cycling test in the range of 100 kHz to 0.1 Hz, with open circuit potential of 0.67 V. Sinusoidal voltage oscillations of 10 mV amplitude were applied.

To measure the ion conductivity of dialysis membrane and Nafion-117 membrane, electrochemical impendence spectroscopy (EIS) measurements were performed at room temperature in the range of 200 kHz to 1 Hz. Sinusoidal voltage oscillations of 10 mV amplitude were applied. The two-electrode configuration setup for ion conductivity tests consists of a sandwich arrangement by placing the 2.0 mol $L^{-1}$ $H_2SO_4/H_2O$ electrolyte-soaked separator membrane sandwiched in between two stainless-steel plates[52]. The ion conductivity of membranes was calculated according to the following equation:

$$\sigma = d/(R \times S) \tag{7}$$

Where $d$ is the thickness of membrane (58.7 μm for MWCO1,000 dialysis membrane and 175.6 μm for Nafion-117 membrane), $S$ is the effective area of membranes (4 $cm^2$), $R$ is the electrical resistance of membrane:

$$R = R_x - R_e \tag{8}$$

Where $R_x$ is the Z'-axis intersection of the EIS curve, $R_e$ (0.873 Ω) is the electrical resistance of the stainless-steel electrode.

**Pretreatment of membranes and electrodes**. The regenerated cellulose-based dialysis membranes severed as separators were cut into squares and soaked in an aqueous solution containing 2 wt.% $NaHCO_3$ and 0.001 mol $L^{-1}$ ethylene diamine tetraacetic acid (EDTA). The solution was heated at 80 °C for ten minutes. After being rinsed with deionized water, the dialysis membranes were soaked in an aqueous solution containing 0.001 mol $L^{-1}$ EDTA and heated at 80 °C for another ten minutes. The pretreated dialysis membranes were stored in deionized water at room temperature. The Nafion-117 membranes were pretreated by successively heating in 5% hydrogen peroxide solution, deionized water, and 1.0 mol $L^{-1}$ sulfuric acid solution at 80 °C for 30 min. The pretreated Nafion-117 membranes were stored in 2.0 mol $L^{-1}$ sulfuric acid solution at room temperature. The carbon paper electrodes were rinsed with anhydrous ethanol and deionized water, and then vacuum dried at 70 °C.

**Battery tests**. Battery tests were carried out using a potentiostat (LAND, CT2001A analyzer) and a standard RFB test cell. The digital photo and detailed configuration of the RFB test cell are presented in Fig. 5b and Supplementary Fig. 7. A piece of dialysis membrane (MWCO1,000) or Nafion-117 membrane with an area of ~2 × 2 $cm^2$ was used to separate anolyte and catholyte in RFBs. The active area is based on the area of carbon paper electrode (1 × 1 $cm^2$). For high-concentration APPSBs tests, 10 mL of catholyte with 1.0 mol $L^{-1}$ PHQ and 10 mL of anolyte with 1.0 mol $L^{-1}$ PI were circulated between the RFB cell and the storage tanks with a peristaltic pump (Longer Pump, BT100-2J) at a flow rate of 30 mL $min^{-1}$. For low-concentration APPSBs tests, the cell based on 0.1 mol $L^{-1}$ PHQ and 0.1 mol $L^{-1}$ PI particulate suspensions were cycled under static operation. The volumetric specific capacity (Ah $L^{-1}$) is based on the total volume of suspension (20 mL).

**Theoretical calculation methods**. All density functional theory (DFT) calculations were performed with the Gaussian 09 program package[53]. Geometrical structures of PI1 unit and its redox intermediates were optimized using the B3LYP/6-31+G (d, p) basis set[54]. The vibrational frequency scaling factor of B3LYP/6-31+G (d, p) basis set is 0.964. UV-Vis absorption spectra of PI1 molecules with single-, double- and triple-units were calculated with Nstates of 30. The calculated Tauc plots of single-, double- and triple-unit PI1 molecules were acquired based on the calculated UV-Vis absorption spectra.

## Data availability

The data that support the findings of this study are available from the corresponding author upon reasonable request.

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

## Acknowledgements

This work was supported by National Key R&D Program of China (2017YFA0208200, 2016YFB0700600, 2015CB659300), Projects of NSFC (21872069, 51761135104, 21573108), Natural Science Foundation of Jiangsu Province (BK20180008, BK20170644), High-Level Entrepreneurial and Innovative Talents Program of Jiangsu Province, and the Fundamental Research Funds for the Central Universities of China.

## Author contributions

Z.J. and W.Y. conceived the idea of this study and designed the experiments. W.Y. performed the sample synthesis, electrochemical measurements, cell tests, data analysis, and theoretical calculations. W.Y. and C.X.W. designed and constructed the RFB system. W.Y., G.Y.Z., L.B.M., Y.R.W., R.P.C., and Y.H. performed the material characterizations. W.Y., J.Q.T., and J.M. did the theoretical calculation analysis. W.Y., C.X.W., G.Y.Z., L.B.M., Y.R.W., R.P.C., Y.H., L.W., T.C., and Z.J. analyzed the data and discussed the results. W.Y. and Z.J. co-wrote and revised the manuscript. Z.J. supervised the project.

## Additional information

**Competing interests:** The authors declare no competing interests.

