## [Peer Review File · Nature Communications]

Reviewers' comments:

Reviewer #1 (Remarks to the Author):

Review for "All-Polymer Particulate Slurry Batteries"

This work describes a redox-flow battery (RFB) in which the redox active materials (for both cathode and anode) are polymer colloids suspended in aqueous acid medium. Due to the discrete nature of the catholyte/anolyte particles (~1 micron diameter cathode particle and 2.7 to 5.5 micron diameter anode particle), the authors could replace Nafion by simple size-exclusion membranes as separators in the flow system. However, the use of polymer colloids as active materials and dialysis membrane as separator based on size-exclusion principle is not new (see similar examples: E. C. Montoto et al., JACS 2016, 138, 13230; T. Janoschka et al. Nature 2015, 527, 78). In addition, the use of electroactive colloids as both catholyte and anolyte were reported in the papers cited above. Though the merits of this work over Janoschka's work were clarified in the manuscript and are judged to be significant, they were already reported by Montoto. In light of the papers cited above, the significance and novelty of this manuscript should be further clarified. For instance, the cycle stability reported in the work is much higher, please explain why relative to Montoto's work. In entirety, the results reported in this manuscript are very interesting, significant, and appealing to a broad audience, but the novelty should be more clearly explained and justified. I hereby recommend a major revision of this manuscript before I can provide further advice on publishing it in this journal. More specific comments are given below.

i. The manuscript claims that the suspension breaks the solubility limit by increasing the concentration $1.6\times$ (from $\sim 0.6 \text{ mol L}^{-1}$ to 1 mol L^{-1} , based on quinone-unit concentration), which theoretically enables higher Ah L^{-1} . However, it appears as though having the active material as particles may limit the capacity. Although the authors mention that the capacity may not come from only the surface of the particles, there is no experiment proving that. In addition, the manuscript reports a combined gravimetric capacity of $74 \text{ mAh per gram of catholyte+anolyte}$, which is substantially lower ($\sim 1/3\times$) than the theoretical combined capacity of catholyte + anolyte ($\sim 200 \text{ mAh g}^{-1}$). The authors should discuss and indicate with calculations what are the performance advantages and limitations of the particle system with the current state-of-the-art organic RFB (i.e. best results in the field). Please include solubility comparisons as well.

ii. Are the PHQ and PI particles porous? What is the stability of these suspensions at low temperature? The authors should also discuss the effect of the particle size on the suspension's electrochemical properties, such as the diffusion coefficients.

iii. What is the reason for the low coulombic efficiency at 5 mA cm^{-2} ? Assuming there is no crossover of material, is there any parasitic reaction happening in this battery?

iv. What is the mechanism for the capacity fading? Although the dialysis membrane keeps the particles in different compartments, the polymers could degrade and release low molecular weight portions during the battery cycles. A careful analysis of the suspensions after battery cycling must be provided to indicate that the particles keep approximately the original dimensions.

v. In Fig. S9, the wavelength of the peaks should be included in the graph. I do not observe any 320 nm peak redshift. The oxidized and reduced species spectra for PHQ and PI seems to have several transition peaks in common. This could indicate partial oxidation/reduction in the particles.

vi. Please clarify the metrics calculation. Is the capacity per L (Ah L^{-1}) based on the total volume of suspension ($\sim 20 \text{ mL}$)? For the current density, what is the area based on?

Reviewer #2 (Remarks to the Author):

This manuscript demonstrated the idea of All-polymer Particulate Slurry Batteries (APPSB), which is similar to the idea of inorganic material particle slurry batteries, e.g. using lithium ion battery materials. The authors claimed the APPSB can break through the solubility limits of active materials and take advantage of the fast kinetics of the redox reactions between the polymer particles and protons. The idea is relatively novel. However, the concept and some of the experimental data need further justification. Major revisions are needed for qualification of being published. My comments and questions are following:

1. First of all, the concept of the all-polymer particulate slurry batteries need further explanation. The authors did not mention how the electricity was conducted within the cell. For example, is it by polymer particles colliding with current collector? The authors need to explain in detail of what are the functions of each component of the cell shown in supplementary figure 7, especially the rolls of titanium current collector, graphite plate and carbon paper electrode. The authors wrote in page 2 line 79, PI is usually regarded as an insulator. I am just confused on how electricity is well conducted at the interface.
2. Despite of the evidence showed by the authors that the redox reactions are very reversible, the electrochemical performance of this slurry battery is in my opinion much worse compared with all vanadium flow battery. Although being a proof-of-concept paper, the authors need to demonstrate the room for further improvement. The authors need to demonstrate the electrochemical performance with a higher particle concentration in the suspensions.
3. The author also need to calculate the energy density beside the specific capacity to give the readers a more comprehensive picture. Also in the manuscript, the authors should compare the performance of this polymer slurry battery with state-of-art flow battery.
4. In page 9 line 271, the coulombic efficiency is 87%. Why is it so low even after the authors claimed there was no side reactions? Also in page 10 line 289, why the coulombic efficiency increases with increasing current densities?

Reviewers' comments:

Reviewer #1 (Remarks to the Author):

Review for "All-Polymer Particulate Slurry Batteries"

This work describes a redox-flow battery (RFB) in which the redox active materials (for both cathode and anode) are polymer colloids suspended in aqueous acid medium. Due to the discrete nature of the catholyte/anolyte particles (~1 micron diameter cathode particle and 2.7 to 5.5 micron diameter anode particle), the authors could replace Nafion by simple size-exclusion membranes as separators in the flow system. However, the use of polymer colloids as active materials and dialysis membrane as separator based on size-exclusion principle is not new (see similar examples: E. C. Montoto et al., JACS 2016, 138, 13230; T. Janoschka et al. Nature 2015, 527, 78). In addition, the use of electroactive colloids as both catholyte and anolyte were reported in the papers cited above. Though the merits of this work over Janoschka's work were clarified in the manuscript and are judged to be significant, they were already reported by Montoto. In light of the papers cited above, the significance and novelty of this manuscript should be further clarified. For instance, the cycle stability reported in the work is much higher, please explain why relative to Montoto's work. In entirety, the results reported in this manuscript are very interesting, significant, and appealing to a broad audience, but the novelty should be more clearly explained and justified. I hereby recommend a major revision of this manuscript before I can provide further advice on publishing it in this journal. More specific comments are given below.

Response: Thanks for your helpful suggestion. In this study, polyhydroquinone (PHQ) and polyimide (PI) are introduced as active materials in RFBs for the first time. Different from incorporating redox molecules as pendants into inactive polymer colloids, we utilized polymers with redox-active sites on their main-chain aromatic rings. Both PHQ and PI can perform rapid multi-electron transfer through polymer chains, and have the merits of large theoretical capacities, easy-to-synthesis, high chemical stability and cheap cost. Additional advantages originate from the robust frameworks of PHQ and PI, as well as the absence of side reactions, allowing excellent charging/discharging cycling stability, which has been demonstrated in other systems, such as Li⁺, Na⁺, and Mg²⁺ ion batteries^{26,28,30,31}. We find that they can be employed as a superb organic redox pair for APPSBs, owing to the high dispersibility, good redox reversibility and suitable electrochemical window of PHQ and PI particulates. Moreover, given the fact that a large number of organic redox-active materials have low solubility in water and the utilization of organic solvents may result in low ion mobility and poor rate capability⁹, the good dispersibility of PHQ and PI particulates in acidic aqueous media is very conducive to construct an aqueous RFB system with high redox reaction kinetics. Compared with the impeded charge transfer in nonaqueous electrolytes, the charge transfer rates in acidic aqueous media are highly improved (6.72×10^{-4} cm/s for PHQ and 2.31×10^{-3} cm/s for PI1, respectively). The energy density and cycling stability of PHQ/PI APPSBs are comparable to the level of other state-of-the-art RFBs. Besides, the molecular structure and particle size of the polymer particulates were demonstrated to influence the electrochemical and physicochemical properties. Further investigation on the improvement of the battery performance by finely tailoring the electrochemical and physicochemical properties of redox-active polymer species for APPSBs will be an important aspect of our future research. We expect this work may provide new insights for exploiting novel redox-active species and

other key components in large-scale energy storage systems.

Here, we have cited the two excellent articles mentioned by the reviewer as Reference [23] and [49]:

23. Janoschka, T. et al. An aqueous, polymer-based redox-flow battery using non-corrosive, safe, and low-cost materials. *Nature* **527**, 78-81 (2015).

49. Montoto, E. C. et al. Redox active colloids as discrete energy storage carriers. *J. Am. Chem. Soc.* **138**, 13230-13237 (2016).

Also, we have adjusted the descriptions of the significance and novelty of our work as you suggested, as follows:

“A new class of RFBs based on combining redox-active polymers or colloids with size-exclusion separator could inhibit the crossover of active species, improve the cycling stability and reduce the battery cost^{23,49}.”

“To alleviate these issues, herein, we propose the design of a new type of RFBs based on aqueous-dispersed all-polymer particulate slurry electrolytes with multi-electron redox capability and fast charge transfer.”

“The application of particulate slurry electrolytes in RFBs makes it possible to replace expensive ion exchange membranes with much cheaper commercial dialysis membranes through the mechanism of size exclusion^{23,49}.”

“Different from incorporating redox molecules as pendants into inactive polymer colloids, we utilized polymers with redox-active sites on their main-chain aromatic rings.”

“In summary, we developed a new type of RFBs by utilizing all-polymer particulate slurries as aqueous catholyte and anolyte.”

i. The manuscript claims that the suspension breaks the solubility limit by increasing the concentration 1.6× (from ~ 0.6 mol L⁻¹ to 1 mol L⁻¹, based on quinone-unit concentration), which theoretically enables higher Ah L⁻¹. However, it appears as though having the active material as particles may limit the capacity. Although the authors mention that the capacity may not come from only the surface of the particles, there is no experiment proving that. In addition, the manuscript reports a combined gravimetric capacity of 74 mAh per gram of catholyte+anolyte, which is substantially lower (~1/3×) than the theoretical combined capacity of catholyte + anolyte (~200 mAh g⁻¹). The authors should discuss and indicate with calculations what are the performance advantages and limitations of the particle system with the current state-of-the-art organic RFB (i.e. best results in the field). Please include solubility comparisons as well.

Response: Thanks for your helpful suggestion. The CV curves of PHQ particulates and benzoquinone monomer (BQ) were compared to demonstrate the site-hopping mechanism of charge transferring during redox processes. PHQ with average diameters of 1 μm and 50 μm are termed as PHQ-1 and

PHQ-50 (Supplementary Figure 7). As shown in Supplementary Figure 8a, with the decrease of particulate size, the electrochemical polarization (ΔE) of PHQ particulates was reduced from 618 mV (PHQ-50) to 200 mV (PHQ-1), indicating the sluggish charge transfer within and among PHQ particulates was improved. At the scan rate of 0.025 V/s, the CV oxidation peak currents of BQ, PHQ-1 and PHQ-50 are 0.74, 0.38 and 0.25 mA/cm², respectively, indicating the large particle size has negative effect to the utilization ratio of redox active species³³. When the scan rate is decreased to 0.006 V/s, the CV oxidation peak currents of BQ, PHQ-1 and PHQ-50 are 0.10, 0.08, 0.06 mA/cm², respectively (Supplementary Figure 8b). These results indicate that the smaller particle size may lead to more charge transfer inside the particulates and more polymer units involved in the redox reaction. Based on the theoretical model proposed in the literatures^{33,49,53}, the site-hopping mechanism is proposed to elucidate the charge transfer of particulates during redox processes. As shown in the new Fig. 1b (see below), the redox-active sites on particulate surface are firstly reduced when approaching to the electrode. Then, the charges transport across the polymer chains by electron hopping between the highly populated redox-active groups^{49,52,53}. In addition to the above site-hopping mechanism, the π -conjugated structures of PHQ and PI also enhance the charge transfer during redox processes. Besides, the strong acidic environment and high H⁺ concentration could also enhance the proton conductivity of polymer particulates, thus accelerating the transfer of electrolyte ions through polymer and increasing the Faradaic response⁵⁴.

We also supplemented the battery performance comparison with the state-of-the-art organic RFBs and discussed the performance advantages and limitations of the APPSBs system. As shown in Supplementary Table 1, the mole concentration of PHQ and PI units for battery testing (1.0 mol/L) in our APPSBs is larger than most of other batteries (usually less than 1.0 mol/L). Benefit from the robust polymer frameworks and the absence of side reactions, the APPSBs in this work exhibits good long-term cycling stability, which is superior to many organic RFBs. Although the rate and capacity utilization of APPSBs are relatively lower than some other aqueous systems, they're still higher than nonaqueous systems. We suggest that the utilization ratio of active materials could be further improved by tuning the size, microstructure and compositions of electrochemical-active particulates, such as constructing conductive agent composites, as well as adding proper electrolyte stabilizer without the compromise of electrochemical performances.

We have added new Fig. 1b, Supplementary Table 1, Supplementary Figure 7 & 8, and some related discussion in the revised Manuscript and Supplementary Data, as following:

“According to the Randles-Sevcik equation, a linear increase of the peak current (i) against the square root of the scan rate ($v^{1/2}$) is observed for PHQ (Fig. 4a & Supplementary Figure 10a) and PI1 (Fig. 4d & Supplementary Figure 10d), respectively, indicating the occurrence of charge diffusion inside the particulates rather than only on the surface.”

“To investigate the charge transfer mechanism of particulates, the electrochemical properties of benzoquinone monomer (BQ) and PHQ particulates with different sizes were compared. PHQ with average diameters of 1 μ m and 50 μ m are termed as PHQ-1 and PHQ-50 (Supplementary Figure 7). As shown in Supplementary Figure 8a, with the decrease of particulate size, the electrochemical polarization (ΔE) of PHQ particulates was reduced from 618 mV (PHQ-50) to 200 mV (PHQ-1),

indicating the sluggish charge transfer within and among PHQ particulates was improved. At the scan rate of 0.025 V/s, the CV oxidation peak currents of BQ, PHQ-1 and PHQ-50 are 0.74, 0.38 and 0.25 mA/cm², respectively, indicating the large particle size has negative effect to the utilization ratio of redox active species³³. When the scan rate is decreased to 0.006 V/s, the CV oxidation peak currents of BQ, PHQ-1 and PHQ-50 are 0.10, 0.08, 0.06 mA/cm², respectively (Supplementary Figure 8b). These results indicate that the smaller particle size may lead to more charge transfer inside the particulates and more units involved in the redox reaction. Based on the theoretical model proposed in the literatures^{33,49,53}, a site-hopping mechanism is proposed to elucidate the charge transfer of particulates during redox processes. As shown in Fig. 1b, the redox-active sites on particulate surface are firstly reduced when approaching to the electrode. Then, the charges transport across the polymer chains by electron hopping between the highly populated redox-active groups^{49,52,53}.”

“Promoted by the good redox kinetics and reversibility of PHQ and PI, the PHQ/PI APPSBs can be stably operated for long-term cycling, delivering capacity of 4.95 Ah/L (3.1 Wh/L) at the current density of 20 mA/cm² with a capacity retention of 70% after 300 cycles, which is comparable to other current state-of-art RFBs (Supplementary Table 1).”

“...as well as adding proper electrolyte stabilizer without the compromise of electrochemical performances.”

33. Xu, C. & Aoki, K. Enhancement of electrochemical activity by small-sizing the vinylferrocene-immobilized polystyrene latex particles. *Langmuir* **20**, 10194-10199 (2004).
50. Yang, Z. G. et al. Electrochemical energy storage for green grid. *Chem. Rev.* **111**, 3577-3613 (2011).
51. Lu, W. J., Shi, D. Q., Zhang, H. M. & Li, X. F. Highly selective core-shell structural membrane with cage-shaped pores for flow battery. *Energy Storage Materials*, **17**, 325-333 (2019).
52. Krause, L. J., Lugg, P. S. & Speckhard, T. A. Electronic conduction in polyimides. *J. Electrochem. Soc.* **136**, 1379-1385 (1989).
53. Oyaizu, K., Kawamoto, T., Suga, T. & Nishide, H. Synthesis and charge transport properties of redox-active nitroxide polyethers with large site density. *Macromolecules* **43**, 10382-10389 (2010).
54. Wang, X. F. et al. Hydronium-ion batteries with perylenetetracarboxylic dianhydride crystals as an electrode. *Angew. Chem. Int. Ed.* **56**, 2909-2913 (2017).

Fig. 1 | Schematic diagram of PHQ/PI1 APPSB. a, Schematic configuration of PHQ/PI1 APPSB. The PHQ/PI1 APPSB mainly consists of two electrolyte reservoirs, two peristaltic pumps and an electrochemical cell where the redox reactions take place. The particulate slurry catholyte and anolyte are separated by commercial dialysis membrane. The electrolytes are circulated between the electrochemical cell and the storage reservoirs during the charging/discharging processes. **b,** Schematic diagram of proposed site-hopping mechanism to elucidate the charge transfer of particulates in the redox processes.

Supplementary Figure 7 | SEM images of (a) PHQ-1 and (b) PHQ-50.

Supplementary Figure 8 | CV curves of 0.005 mol/L BQ, PHQ-1 and PHQ-50 in 2.0 mol/L H₂SO₄ aqueous solution at the scan rate of (a) 0.025 V/s and (b) 0.006 V/s.

Supplementary Table 1 | Performance comparisons of the APPSBs in this work with other representative vanadium-based and organic-based RFBs.

Sources	Electrolyte	Redox-active material	Electron-transfer rate constant k_0 (cm/s)	Concentration (mol/L)	Current density (mA/cm ²)	Energy density (Wh/L)	Cycle number	Capacity retention
This work	H ₂ SO ₄ (aq)		6.72×10^{-4}	0.1, 1.0	5-20	2-6	300	70%
			2.31×10^{-3}	0.1, 1.0			5,000 (non-flow cell)	74%
			1.01×10^{-4}	0.1, 1.0				
Ref. 45, 50, 51	H ₂ SO ₄ (aq)	V ³⁺ /V ²⁺	5.3×10^{-4}	2	200	10-20	5,000	—
		VO ²⁺ /VO ₂ ⁺	2.8×10^{-6}					
Ref. 23	NaCl (aq)		4.5×10^{-4}	0.07, 0.37	20	3.6	10,000 (non-flow cell)	80%
			9×10^{-4}	0.15, 0.37	40	10	100	67%

Ref. 22	KCl (aq)		5.3×10^{-3}	0.24 (Maximum: 1.5)	80	4.8	100	99%
Ref. 21	KOH (aq)		1.2×10^{-5}	0.5 (maximum: 2)	100	12	400	91%
Ref. 24	NaCl (aq)		2.2×10^{-2}	0.75, 1.3	50	5	500	97%
			1.4×10^{-2}	1, 1.3	50	9.6	250	99%
Ref. 46	0.1 mol/L TBAPF ₆ in acetonitrile		1.42×10^{-2}	0.1 (maximum 3.8)	1	4.1	20	60%
Ref. 20	1.0 mol/L LiTFSI in DME		2.46×10^{-3}	0.3, 0.3	35	6.5	50	95%
			1.35×10^{-2}	0.1, 0.1	10	2.3	100	85%
Ref. 47	NaCl (aq)		3.66×10^{-5}	0.5, 0.5 (maximum: 3)	60	9	700	91%
			4.60×10^{-6}	0.7, 0.7 (maximum: 2)	60	9.9	500	81%
Ref. 48	NaCl (aq)		2.8×10^{-4}	0.5, 0.5 (maximum: 3)	60	7.5	100	89%
			2.6×10^{-4}	0.5				
Ref. 17	H ₂ SO ₄ (aq)		7.2×10^{-3}	0.1, 1	200	1	15	99%
		Br ₂	—	0.5, 0.5	500	16	10	92%
Ref. 49	0.1 mol/L LiBF ₄ in acetonitrile	Viologen-based colloids	1.5×10^{-2}	0.01 (maximum: 2)	0.043	0.05	50	100%
		Ferrocene-based colloids	—	0.01				

ii. Are the PHQ and PI particles porous? What is the stability of these suspensions at low temperature? The authors should also discuss the effect of the particle size on the suspension's electrochemical properties, such as the diffusion coefficients.

Response: Thanks very much for your valuable questions. We have investigated the porosity characteristics of PHQ and PI particulates by N₂ adsorption–desorption isotherms. The Brunauer–Emmer–Teller (BET) surface areas of PHQ, PI1 and PI2 particulates are 5.5, 19.9, 2.0 m²/g, respectively (Supplementary Figure 19). The pore size distribution in Supplementary Figure 19b demonstrates the presence of some mesopores in PI1 particulates, although the pore volume is relatively low.

At low temperature (4 °C), visual observations of the dispersion stability of polymer particulate slurries only show slight sedimentation after three days (see the new Supplementary Figure 1b, as below). Besides, the electrochemical tests of APPSBs were normally performed with the rapid circulation of electrolytes driven by peristaltic pumps, so the precipitation of polymer particulates was very minimal under the flow mode, even at low temperature.

Following your valuable suggestions, we have compared the dispersibility, viscosity, and electrochemical properties of polymer particulates with different sizes. The particulate sizes of PI were decreased to <1 μm via ball-milling method. CV tests and RDE measurements show the smaller PI particulates possess better charge transfer and higher capacity utilization (see the new Supplementary Figure 24–26 below). Detailed comparison is listed in the new Supplementary Table 2.

We have added the following discussion and figures in the revised Manuscript and Supplementary Data, as below:

“Nitrogen adsorption/desorption isotherms were recorded at 77 K using a Micrometrics ASAP 2020 analyzer. The surface area and pore size distribution were determined by Brunauer–Emmett–Teller (BET) and Barrett–Joyner–Halenda method, respectively.”

“Significant difference of redox kinetics of PI1 and PI2 particulates may be partially originated from their different pore structures. Brunauer–Emmer–Teller (BET) analysis shows the specific surface areas of 19.9 and 2.0 m²/g for PI1 and PI2 particulates, respectively (Supplementary Figure 20). The pore size distribution demonstrates the presence of some mesopores of PI1, although the pore volume is relatively low. Compared to the compact stacking structure of PI2 resulted from the rigid benzene rings, the loosely-stacking structure of PI1 attributed by the flexible alkyl segments on the main chains may accelerate the transfer of abundant electrolyte ions into the particulates.”

“...at room temperature (25 °C) and low temperature (4 °C) ...”

“The influences of particle size on the electrochemical and physicochemical properties of polymer particulates were investigated. Briefly, ballmilling processes were performed to further decrease the size of PI1 and PI2 particulates, and the control samples after ballmilling for 48 h were termed as PI1-ballmilled, PI2-ballmilled, respectively. As shown in the SEM images and DLS curves (Supplementary Figure 22 & 23), the average particle size of PI1 decreased from 2.7 μm (PI1) to 0.8 μm (PI1-ballmilled), and the average particle size of PI2 decreased from 5.6 μm (PI2) to 0.9 μm (PI2-ballmilled). The Zeta potentials of PI1-ballmilled and PI2-ballmilled were measured to be 47.8 mV and 34.3 mV,

indicating the dispersibility and stability were improved after ball-milling. CV analysis revealed the increased Faradaic response of the smaller PI1 and PI2 (Supplementary Figure 24). Diffusion coefficient, including the physical transport of particulates to the electrode and the charge transport of the particulates, are calculated to be $1.7 \times 10^{-7} \text{ cm}^2/\text{s}$ (PI1), $3.4 \times 10^{-7} \text{ cm}^2/\text{s}$ (PI1-ballmilled), $0.7 \times 10^{-7} \text{ cm}^2/\text{s}$ (PI2), and $1.3 \times 10^{-7} \text{ cm}^2/\text{s}$ (PI2-ballmilled) (Supplementary Figure 10, 18, 25, 26). The smaller particle size accelerates the particulate diffusion and charge transport in the redox process (Supplementary Table 2). Constant-current charge/discharge tests of PHQ/PI2-ballmilled APPSB shows discharging capacity of 8.40 Ah/L (6.05 Wh/L) at the current density of $5 \text{ mA}/\text{cm}^2$, larger than that of PHQ/PI2 APPSB (4.30 Ah/L, 3.02 Wh/L), demonstrating higher capacity utilization of smaller polymer particulates (Supplementary Figure 27). We suggest that the utilization ratio of active materials could be further improved by tuning the size, microstructure and compositions of electrochemical-active particulates, such as constructing conductive agent composites, as well as adding proper electrolyte stabilizer without the compromise of electrochemical performances.”

Supplementary Figure 1 | Visual observations of the dispersion stability of polymer particulate slurries at (a) 25 °C and (b) 4 °C.

Supplementary Figure 19 | N₂ adsorption–desorption isotherms of (a) PI1, (c) PHQ and (d) PI2 particulates, respectively. (b) Pore size distribution of PI1 particulates.

Supplementary Figure 22 | SEM images of PI with different sizes: (a) PI1, (b) PI1-ballmilled, (c) PI2, and (d) PI2-ballmilled.

Supplementary Figure 23 | Size distribution and dispersion stability of ballmilled PI particulate suspensions. DLS diameter distributions of (a) PI1-ballmilled and (b) PI2-ballmilled. Zeta potential curves of (c) PI1-ballmilled and (d) PI2-ballmilled.

Supplementary Figure 24 | CV curves of 0.005 mol/L PI particulate suspensions in 2.0 mol/L H₂SO₄ aqueous solution. (a) PI1 and PI1-ballmilled, (b) PI2 and PI2-ballmilled. The scan rate is 0.025 V/s.

Supplementary Figure 25 | Electrochemical characterizations of P11-ballmilled. (a) Cyclic voltammograms of 0.1 mol/L P11-ballmilled in 2.0 mol/L H₂SO₄ aqueous solution at different scan rates. (b) Oxidation peak current (*i*) versus scan speed. (c) RDE measurements at rotating electrode speeds from 400 rpm to 3,600 rpm using 0.005 mol/L P11-ballmilled in 2.0 mol/L H₂SO₄ aqueous solution. (d) Limiting current versus the square root of rotation velocity (Levich plot). (e) Koutecky-Levich plot. (f) Tafel plot.

Supplementary Figure 26 | Electrochemical characterizations of PI2-ballmilled. (a) Cyclic voltammograms of 0.1 mol/L PI2-ballmilled in 2.0 mol/L H₂SO₄ aqueous solution at different scan rates. **(b)** Oxidation peak current (*i*) versus scan speed. **(c)** RDE measurements at rotating electrode speeds from 400 rpm to 3,600 rpm using 0.005 mol/L PI2-ballmilled in 2.0 mol/L H₂SO₄ aqueous solution. **(d)** Limiting current versus the square root of rotation velocity (Levich plot). **(e)** Koutecky-Levich plot. **(f)** Tafel plot.

Supplementary Figure 27 | Performances of PHQ/PI2-ballmilled APPSBs. (a) Representative charge/discharge curves of PHQ/PI2-ballmilled APPSBs at 5 mA/cm². (b) Stability test of PHQ/PI2-ballmilled APPSBs at the current density of 5 mA/cm².

Supplementary Table 2. Physicochemical and electrochemical properties of polymer particulates with different particle sizes.

Materials	Average size (μm)		Zeta potential (mV)	Oxidation peak current (mA/cm ²)	Polarization (mV)	Diffusion Coefficient (10 ⁻⁷ cm ² /s)	Electron transfer rate constant (10 ⁻³ cm/s)	Viscosity (mPa s)
	SEM	DLS						
PI1	2.5	2.7	48.6	0.42	38	1.7	2.31	3.1
PI1-ballmilled	0.8	0.9	47.8	0.50	30	3.4	1.33	5.2
PI2	5.6	5.6	10.6	0.07	91	0.7	0.10	2.3
PI2-ballmilled	0.9	0.9	34.3	0.32	57	1.3	0.81	4.7

iii. What is the reason for the low coulombic efficiency at 5 mA cm⁻²? Assuming there is no crossover of material, is there any parasitic reaction happening in this battery?

Response: Thanks very much for the good questions. As shown in the new Supplementary Figure 4, we noticed that the CV curve of PHQ particulate suspension at the 50th cycle was nearly overlapped with that at the 1st and 2nd cycles, while the ratio of oxidation capacity to reduction capacity (Q₁/Q₂) were changed at different scan rates. The Q₁/Q₂ is 1.08 at 0.1 V/s and 1.34 at 0.025 V/s, respectively. We attributed the small oxidation peak at 1.2 V (vs. SHE) to the electro-polymerization of PHQ^{55,56}. As shown in Supplementary Figure 5, when the terminal hydroquinone was oxidized to the protonated benzoquinone in the electro-oxidation process, it might react with the non-protonated hydroquinone, leading to the electro-polymerization of polymer chains. The electro-polymerization provides additional oxidation capacity, but it doesn't affect the reversibility of redox-active groups. The parasitic reaction of electro-polymerization can be suppressed by increasing the charge rate, as indicated by the Tafel plots (Supplementary Figure 6).

We have added the related discussion and figures in the revised Manuscript and Supplementary data, as below:

“As shown in Supplementary Figure 4, the CV curve of PHQ particulate suspension at the 50th cycle was nearly overlapped with that at the 1st and 2nd cycles, while the ratio of oxidation capacity to reduction capacity (Q_1/Q_2) were changed at different scan rates. The Q_1/Q_2 is 1.08 at 0.1 V/s and 1.34 at 0.025 V/s, respectively. The small oxidation peak at 1.2 V (vs. SHE) is proposed to be originated from the electro-polymerization of PHQ^{55,56}. As illustrated in Supplementary Figure 5, when the terminal hydroquinone was oxidized to the protonated benzoquinone in the electro-oxidation process, it might react with the non-protonated hydroquinone, leading to the electro-polymerization of polymer chains. The electro-polymerization can provide additional oxidation capacity, but it doesn't affect the reversibility of redox-active groups. The parasitic reaction of electro-polymerization can be suppressed by increasing the current rate, as indicated by the Tafel plots (Supplementary Figure 6).”

“The increasing Coulombic efficiency with increasing current densities may be attributed by the aforementioned suppression of the parasitic electro-polymerization reaction of PHQ at high current densities (Supplementary Figure 5 & 6).”

55. Yamamoto, K., Asada, T., Nishide, H. & Tsuchida, E. The preparation of poly (dihydroxyphenylene) through the electro-oxidative polymerization of hydroquinone. *B. Chem. Soc. Jpn.* **63**, 1211-1216 (1990).

56. Wang, P., Martin, B. D., Parida, S., Rethwisch, D. G. & Dordick, J. S. Multienzymic synthesis of poly(hydroquinone) for use as a redox polymer. *J. Am. Chem. Soc.* **117**, 12885-12886 (1995).

Supplementary Figure 4 | CV curves of 0.005 mol/L PHQ particulates in 2.0 mol/L H₂SO₄ aqueous solution during the 1st, 2nd and 50th cycles at different scan rate of (a) 0.1 V/s and (b) 0.025 V/s, within a potential range from 0 to 1.5 V vs. SHE.

Supplementary Figure 5 | Proposed electro-polymerization process of PHQ in acidic solution.

Supplementary Figure 6 | Tafel plots of proposed parallel reactions of electro-polymerization and electro-oxidation of PHQ.

iv. What is the mechanism for the capacity fading? Although the dialysis membrane keeps the particles in different compartments, the polymers could degrade and release low molecular weight portions during the battery cycles. A careful analysis of the suspensions after battery cycling must be provided to indicate that the particles keep approximately the original dimensions.

Response: Thanks very much for the good questions. The capacity fading may be attributed to the aggregation and sedimentation of active particulates after long-time cycling. DLS and Zeta potential measurements reveal the similar dispersibility and stability of the diluted polymer particulate suspensions after cycling test (Supplementary Figure 13). As shown in the SEM images (Supplementary Figure 14), most of the PHQ and PI1 particulates keep approximately the original dimensions, but some particulates aggregate into larger particles. For PI2, the aggregation is much worse. The agglomerates may precipitate in the flow grooves and reservoirs, resulting in the capacity fading. Therefore, to further improve the cycling stability of APPSBs, we suggest that proper particulate stabilizer without the compromise of electrochemical performance could be introduced.

 “DLS and Zeta potential measurements reveal the similar dispersibility and stability of the diluted polymer particulate suspensions after cycling test (Supplementary Figure 13). As shown in the SEM images (Supplementary Figure 14), most of the PHQ and PI1 particulates keep approximately the original dimensions, but some particulates aggregate into larger particles. For PI2, the aggregation is much worse. The agglomerates may precipitate in the flow grooves and reservoirs, resulting in the capacity fading. Therefore, to further improve the cycling stability of APPSBs, we suggest that proper particulate stabilizer without the compromise of electrochemical performance could be introduced.”

Supplementary Figure 13 | DLS distribution and Zeta potential of (a, c) PHQ and (b, d) PI1 particulate suspensions after charging/discharging cycling, respectively.

Supplementary Figure 14 | SEM images of (a) PHQ, (b) PI1, and (c) PI2 particulates after charging/discharging cycling.

v. In Fig. S9, the wavelength of the peaks should be included in the graph. I do not observe any 320 nm peak redshift. The oxidized and reduced species spectra for PHQ and PI seems to have several transition peaks in common. This could indicate partial oxidation/reduction in the particles.

Response: Thanks very much for the good question. The accurate wavelengths of the UV-Vis absorption peaks have been supplemented in Supplementary Figure 15, and more discussion has been added in the revised Manuscript, as following:

Supplementary Figure 15 | UV-Vis absorption spectra of the diluted polymer particulate suspensions at fully charged and discharged states. The polymer particulate suspensions at fully-discharged state represent the reduced form of PHQ and the oxidized form of P11. The polymer particulate suspensions at fully-charged state represent the oxidized form of PHQ and the reduced form of P11.

“After the electro-oxidation of PHQ, a new absorption peak at 246 nm was emerged (Supplementary Figure 15a), and the absorption peak at 326 nm was enhanced and red-shifted, indicating the conjugation effect between the carbonyl groups and the backbone of benzene rings³⁶. On the other hand, as shown in Supplementary Figure 15b, the electro-reduction of P11 leads to new absorption peaks at 304 nm, 539 nm and 645 nm³⁷, owing to the spatial charge distribution variation of π -conjugation system.”

vi. Please clarify the metrics calculation. Is the capacity per L (Ah L⁻¹) based on the total volume of suspension (~ 20 mL)? For the current density, what is the area based on?

Response: Thanks. The capacity per L (Ah L⁻¹) is based on the total volume of suspension (~ 20 mL). The area is based on the area of carbon paper electrode (1×1 cm²).

“The volumetric specific capacity (Ah L⁻¹) is based on the total volume of suspension (20 mL).”

“The active area is based on the area of carbon paper electrode (1×1 cm²).”

Reviewer #2 (Remarks to the Author):

This manuscript demonstrated the idea of All-polymer Particulate Slurry Batteries (APPSB), which is similar to the idea of inorganic material particle slurry batteries, e.g. using lithium ion battery materials. The authors claimed the APPSB can break through the solubility limits of active materials and take advantage of the fast kinetic of the redox reactions between the polymer particles and protons. The idea is relatively novel. However, the concept and some of the experimental data need further justification. Major revisions are needed for qualification of being published. My comments and questions are following:

1. First of all, the concept of the all-polymer particulate slurry batteries need further explanation. The authors did not mention how the electricity was conducted within the cell. For example, is it by polymer particles colliding with current collector? The authors need to explain in detail of what are the functions of each component of the cell shown in supplementary figure 7, especially the rolls of titanium current collector, graphite plate and carbon paper electrode. The authors wrote in page 2 line 79, PI is usually regarded as an insulator. I am just confused on how electricity is well conducted at the interface.

Response: Thanks very much for the helpful suggestions. Based on the theoretical model proposed in the literatures^{33,49,52}, the CV curves of PHQ particulates and benzoquinone monomer (BQ) were compared to demonstrate the site-hopping mechanism of charge transferring during redox processes. As shown in the new Fig. 1b (see below), the redox-active sites on the surface of PHQ and PI particulates are firstly reduced when approaching to the electrode. Then, the charges transport across the polymer chains by electron hopping between the highly populated redox-active groups^{49,52,53}. In addition to the above site-hopping mechanism, the π -conjugated structures of PHQ and PI also enhance the charge transfer during redox processes. Besides, the strong acidic environment and high H⁺ concentration could also enhance the proton conductivity of polymer particulates, thus accelerating the transfer of electrolyte ions through polymer and increasing the Faradaic response⁵⁴.

We also added the functions of each component of the cell in the caption of Supplementary Figure 11. The titanium plate acts as a highly corrosion-resistant current collector. The graphite plate with narrow grooves provides the flow channel of the circulating electrolyte, and the carbon paper electrode provides the electrochemical reaction sites for particulates.

The PI is normally regarded as the electron-insulator, but it can transport charges by site hopping mechanism during redox processes^{49,52,53}, whereby the electrons can transfer from one redox-active imide group to another on a neighboring unit.

We have added the related discussion and figures in the revised Manuscript and Supplementary Data, as follows:

“According to the Randles-Sevcik equation, a linear increase of the peak current (*i*) against the square root of the scan rate ($v^{1/2}$) is observed for PHQ (Fig. 4a & Supplementary Figure 10a) and PI1 (Fig. 4d & Supplementary Figure 10d), respectively, indicating the occurrence of charge diffusion inside the particulates rather than only on the surface.

“To investigate the charge transfer mechanism of particulates, the electrochemical properties of benzoquinone monomer (BQ) and PHQ particulates with different sizes were compared. PHQ with average diameters of 1 μm and 50 μm are termed as PHQ-1 and PHQ-50 (Supplementary Figure 7). As shown in Supplementary Figure 8a, with the decrease of particulate size, the electrochemical polarization (ΔE) of PHQ particulates was reduced from 618 mV (PHQ-50) to 200 mV (PHQ-1), indicating the sluggish charge transfer within and among PHQ particulates was improved. At the scan rate of 0.025 V/s, the CV oxidation peak currents of BQ, PHQ-1 and PHQ-50 are 0.74, 0.38 and 0.25 mA/cm², respectively, indicating the large particle size has negative effect to the utilization ratio of redox active species³³. When the scan rate is decreased to 0.006 V/s, the CV oxidation peak currents

of BQ, PHQ-1 and PHQ-50 are 0.10, 0.08, 0.06 mA/cm², respectively (Supplementary Figure 8b). These results indicate that the smaller particle size may lead to more charge transfer inside the particulates and more units involved in the redox reaction. Based on the theoretical model proposed in the literatures^{33,49,53}, a site-hopping mechanism is proposed to elucidate the charge transfer of particulates during redox processes. As shown in Fig. 1b, the redox-active sites on particulate surface are firstly reduced when approaching to the electrode. Then, the charges transport across the polymer chains by electron hopping between the highly populated redox-active groups^{49,52,53}.

1. Stainless steel plate
2. PTFE plate
3. Titanium current collector
4. Graphite plate
5. Silicone rubber seal
6. PTFE frame
7. Carbon paper electrode
8. Separator
9. Silicone rubber tube

Supplementary Figure 11 | Structure configuration and components of APPSB cell. The stainless-steel plate fixed with screws works as the frame of the cell. The PTFE plate prevents the short-circuit of the cell. The titanium plate acts as a highly corrosion-resistant current collector. The graphite plate with narrow grooves provides the flow channel of the circulating electrolyte, and the carbon paper electrode provides the electrochemical reaction sites for particulates. The silicone rubber seal prevents the leakage of electrolyte. The PTFE frame connecting with silicone rubber tubes is the gateway for electrolytes pumping into and out of the cell. The separator keeps the two electrodes apart to prevent electrical short-circuit while allowing the transport of ions during charge/discharge processes.

52. Krause, L. J., Lugg, P. S. & Speckhard, T. A. Electronic conduction in polyimides. *J. Electrochem. Soc.* **136**, 1379-1385 (1989).

53. Oyaizu, K., Kawamoto, T., Suga, T. & Nishide, H. Synthesis and charge transport properties of redox-active nitroxide polyethers with large site density. *Macromolecules* **43**, 10382-10389 (2010).

54. Wang, X. F. et al. Hydronium-ion batteries with perylenetetracarboxylic dianhydride crystals as an electrode. *Angew. Chem. Int. Ed.* **56**, 2909-2913 (2017).

Fig. 1 | Schematic diagram of PHQ/PI1 APPSB. a, Schematic configuration of PHQ/PI1 APPSB. The PHQ/PI1 APPSB mainly consists of two electrolyte reservoirs, two peristaltic pumps and an electrochemical cell where the redox reactions take place. The particulate slurry catholyte and anolyte are separated by commercial dialysis membrane. The electrolytes are circulated between the electrochemical cell and the storage reservoirs during the charging/discharging processes. **b,** Schematic diagram of proposed site-hopping mechanism to elucidate the charge transfer of particulates in the redox processes.

Supplementary Figure 7 | SEM images of (a) PHQ-1 and (b) PHQ-50.

Supplementary Figure 8 | CV curves of 0.005 mol/L BQ, PHQ-1 and PHQ-50 in 2.0 mol/L H₂SO₄ aqueous solution at the scan rate of (a) 0.025 V/s and (b) 0.006 V/s.

2. Despite of the evidence showed by the authors that the redox reactions are very reversible, the electrochemical performance of this slurry battery is in my opinion much worse compared with all vanadium flow battery. Although being a proof-of-concept paper, the authors need to demonstrate the room for further improvement. The authors need to demonstrate the electrochemical performance with a higher particle concentration in the suspensions.

Response: Thanks very much for your good suggestion. To demonstrate the improved volumetric specific capacity at higher concentration of redox-active units, the particulate sizes of PI1 and PI2 were further decreased to <1 μm via ballmilling method, and these control samples were denoted as PI1-ballmilled and PI2-ballmilled, respectively. The CV tests and RDE measurements of PI1-ballmilled and PI2-ballmilled samples show that the smaller polymer particulates possess better charge transfer and higher capacity utilization (Supplementary Figure 23-26 and Supplementary Table 2). Briefly, constant-current charge/discharge tests of PHQ/PI2-ballmilled APPSB shows discharging capacity of 8.40 Ah/L at 5 mA/cm², which is larger than that of original PHQ/PI2 APPSB (4.30 Ah/L). This result confirms that it is possible to further improve the volumetric energy density of APPSBs with a higher concentration of redox-active species. We suggest that the utilization ratio of active materials could be further improved by tuning the size, microstructure and compositions of electrochemical-active particulates, such as constructing conductive agent composites, as well as adding proper electrolyte stabilizer without the compromise of electrochemical performances.

We have added the related discussion and figures in the revised Manuscript and Supplementary Data, as follows:

“The influences of particle size on the electrochemical and physicochemical properties of polymer particulates were investigated. Briefly, ballmilling processes were performed to further decrease the size of PI1 and PI2 particulates, and the control samples after ballmilling for 48 h were termed as PI1-ballmilled, PI2-ballmilled, respectively. As shown in the SEM images and DLS curves (Supplementary Figure 22 & 23), the average particle size of PI1 decreased from 2.7 μm (PI1) to 0.8 μm (PI1-ballmilled), and the average particle size of PI2 decreased from 5.6 μm (PI2) to 0.9 μm (PI2-ballmilled). The Zeta potentials of PI1-ballmilled and PI2-ballmilled were measured to be 47.8 mV and 34.3 mV, indicating the dispersibility and stability were improved after ball-milling. CV analysis revealed the increased Faradaic response of the smaller PI1 and PI2 (Supplementary Figure 24). Diffusion coefficient, including the physical transport of particulates to the electrode and the charge transport of the particulates, are calculated to be $1.7 \times 10^{-7} \text{ cm}^2/\text{s}$ (PI1), $3.4 \times 10^{-7} \text{ cm}^2/\text{s}$ (PI1-ballmilled), $0.7 \times 10^{-7} \text{ cm}^2/\text{s}$ (PI2), and $1.3 \times 10^{-7} \text{ cm}^2/\text{s}$ (PI2-ballmilled) (Supplementary Figure 10, 18, 25, 26). The smaller particle size accelerates the particulate diffusion and charge transport in the redox process (Supplementary Table 2). Constant-current charge/discharge tests of PHQ/PI2-ballmilled APPSB shows discharging capacity of 8.40 Ah/L (6.05 Wh/L) at the current density of 5 mA/cm², larger than that of PHQ/PI2 APPSB (4.30 Ah/L, 3.02 Wh/L), demonstrating higher capacity utilization of smaller polymer particulates (Supplementary Figure 27). We suggest that the utilization ratio of active materials could be further improved by tuning the size, microstructure and compositions of

electrochemical-active particulates, such as constructing conductive agent composites, as well as adding proper electrolyte stabilizer without the compromise of electrochemical performances.”

Supplementary Figure 22 | SEM images of PI with different sizes: (a) PI1, (b) PI1-ballmilled, (c) PI2, and (d) PI2-ballmilled.

Supplementary Figure 23 | Size distribution and dispersion stability of ballmilled PI particulate suspensions. DLS diameter distributions of (a) PI1-ballmilled and (b) PI2-ballmilled. Zeta potential curves of (c) PI1-ballmilled and (d) PI2-ballmilled.

Supplementary Figure 24 | CV curves of 0.005 mol/L PI particulate suspensions in 2.0 mol/L H₂SO₄ aqueous solution. (a) PI1 and PI1-ballmilled, (b) PI2 and PI2-ballmilled. The scan rate is 0.025 V/s.

Supplementary Figure 25 | Electrochemical characterizations of P11-ballmilled. (a) Cyclic voltammograms of 0.1 mol/L P11-ballmilled in 2.0 mol/L H₂SO₄ aqueous solution at different scan rates. (b) Oxidation peak current (i) versus scan speed. (c) RDE measurements at rotating electrode speeds from 400 rpm to 3,600 rpm using 0.005 mol/L P11-ballmilled in 2.0 mol/L H₂SO₄ aqueous solution. (d) Limiting current versus the square root of rotation velocity (Levich plot). (e) Koutecky-Levich plot. (f) Tafel plot.

Supplementary Figure 26 | Electrochemical characterizations of PI2-ballmilled. (a) Cyclic voltammograms of 0.1 mol/L PI2-ballmilled in 2.0 mol/L H₂SO₄ aqueous solution at different scan rates. **(b)** Oxidation peak current (*i*) versus scan speed. **(c)** RDE measurements at rotating electrode speeds from 400 rpm to 3,600 rpm using 0.005 mol/L PI2-ballmilled in 2.0 mol/L H₂SO₄ aqueous solution. **(d)** Limiting current versus the square root of rotation velocity (Levich plot). **(e)** Koutecky-Levich plot. **(f)** Tafel plot.

Supplementary Figure 27 | Performances of PHQ/PI2-ballmilled APPSBs. (a) Representative charge/discharge curves of PHQ/PI2-ballmilled APPSBs at 5 mA/cm². (b) Stability test of PHQ/PI2-ballmilled APPSBs at the current density of 5 mA/cm².

Supplementary Table 2. Physicochemical and electrochemical properties of polymer particulates with different particle sizes.

Materials	Average size (μm)		Zeta potential (mV)	Oxidation peak current (mA/cm ²)	Polarization (mV)	Diffusion Coefficient (10 ⁻⁷ cm ² /s)	Electron transfer rate constant (10 ⁻³ cm/s)	Viscosity (mPa s)
	SEM	DLS						
PI1	2.5	2.7	48.6	0.42	38	1.7	2.31	3.1
PI1-ballmilled	0.8	0.9	47.8	0.50	30	3.4	1.33	5.2
PI2	5.6	5.6	10.6	0.07	91	0.7	0.10	2.3
PI2-ballmilled	0.9	0.9	34.3	0.32	57	1.3	0.81	4.7

3. The author also need to calculate the energy density beside the specific capacity to give the readers a more comprehensive picture. Also in the manuscript, the authors should compare the performance of this polymer slurry battery with state-of-art flow battery.

Response: Thanks for your helpful suggestion. We have added the energy density beside the specific capacity and the performance comparison with the current state-of-the art organic RFBs. Aqueous and organic electrolytes in the current state-of-art organic RFB are both included in the comparison. As shown in Supplementary Table 1, the mole concentration of PHQ and PI units for battery testing (1.0 mol/L) in our APPSBs is larger than most of other batteries (usually less than 1.0 mol/L). Benefit from the robust polymer frameworks and the absence of side reactions, the APPSBs in this work exhibits good long-term cycling stability, which is superior to many organic RFBs. Although the rate and capacity utilization of APPSBs are relatively lower than some other aqueous systems, they're still higher than nonaqueous systems. We suggest that the utilization ratio of active materials could be further improved by tuning the size, microstructure and compositions of electrochemical-active particulates, such as constructing conductive agent composites, as well as adding proper electrolyte stabilizer without the compromise of electrochemical performances.

“Promoted by the good redox kinetics and reversibility of PHQ and PI, the PHQ/PI APPSBs can be stably operated for long-term cycling, delivering capacity of 4.95 Ah/L (3.1 Wh/L) at the current density of 20 mA/cm² with a capacity retention of 70% after 300 cycles, which is comparable to other current state-of-art RFBs (Supplementary Table 1).”

51. Lu, W. J., Shi, D. Q., Zhang, H. M. & Li, X. F. Highly selective core-shell structural membrane with cage-shaped pores for flow battery. *Energy Storage Materials*, **17**, 325-333 (2019).

52. Krause, L. J., Lugg, P. S. & Speckhard, T. A. Electronic conduction in polyimides. *J. Electrochem. Soc.* **136**, 1379-1385 (1989).

Supplementary Table 1 | Performance comparisons of the APPSBs in this work with other representative vanadium-based and organic-based RFBs.

Sources	Electrolyte	Redox-active material	Electron-transfer rate constant k_0 (cm/s)	Concentration (mol/L)	Current density (mA/cm ²)	Energy density (Wh/L)	Cycle number	Capacity retention
This work	H ₂ SO ₄ (aq)		6.72×10^{-4}	0.1, 1.0	5-20	2-6	300	70%
			2.31×10^{-3}	0.1, 1.0			5,000 (non-flow cell)	74%
			1.01×10^{-4}	0.1, 1.0				
Ref. 45, 50, 51	H ₂ SO ₄ (aq)	V ³⁺ /V ²⁺	5.3×10^{-4}	2	200	10-20	5,000	—
		VO ²⁺ /VO ₂ ⁺	2.8×10^{-6}					
Ref. 23	NaCl (aq)		4.5×10^{-4}	0.07, 0.37	20	3.6	10,000 (non-flow cell)	80%
			9×10^{-4}	0.15, 0.37	40	10	100	67%
Ref. 22	KCl (aq)		5.3×10^{-3}	0.24 (Maximum: 1.5)	80	4.8	100	99%

Ref. 21	KOH (aq)		1.2×10^{-5}	0.5 (maximum: 2)	100	12	400	91%
Ref. 24	NaCl (aq)		2.2×10^{-2}	0.75, 1.3	50	5	500	97%
			1.4×10^{-2}	1, 1.3	50	9.6	250	99%
Ref. 46	0.1 mol/L TBAPF ₆ in acetonitrile		1.42×10^{-2}	0.1 (maximum 3.8)	1	4.1	20	60%
Ref. 20	1.0 mol/L LiTFSI in DME		2.46×10^{-3}	0.3, 0.3	35	6.5	50	95%
			1.35×10^{-2}	0.1, 0.1	10	2.3	100	85%
Ref. 47	NaCl (aq)		3.66×10^{-5}	0.5, 0.5 (maximum: 3)	60	9	700	91%
			4.60×10^{-6}	0.7, 0.7 (maximum: 2)	60	9.9	500	81%
Ref. 48	NaCl (aq)		2.8×10^{-4}	0.5, 0.5 (maximum: 3)	60	7.5	100	89%
			2.6×10^{-4}	0.5				
Ref. 17	H ₂ SO ₄ (aq)		7.2×10^{-3}	0.1, 1	200	1	15	99%
		Br ₂	—	0.5, 0.5	500	16	10	92%
Ref. 49	0.1 mol/L LiBF ₄ in acetonitrile	Viologen-based colloids	1.5×10^{-2}	0.01 (maximum: 2)	0.043	0.05	50	100%
		Ferrocene-based colloids	—	0.01				

4. In page 9 line 271, the coulombic efficiency is 87%. Why is it so low even after the authors claimed there was no side reactions? Also in page 10 line 289, why the coulombic efficiency increases with increasing current densities?

Response: Thanks very much for the good question. As shown in the new Supplementary Figure 4, we noticed that the CV curve of PHQ particulate suspension at the 50th cycle was nearly overlapped with that at the 1st and 2nd cycles, while the ratio of oxidation capacity to reduction capacity (Q_1/Q_2)

were changed at different scan rates. The Q_1/Q_2 is 1.08 at 0.1 V/s and 1.34 at 0.025 V/s, respectively. We attributed the small oxidation peak at 1.2 V (vs. SHE) to the electro-polymerization of PHQ^{55,56}. As shown in Supplementary Figure 5, when the terminal hydroquinone was oxidized to the protonated benzoquinone in the electro-oxidation process, it might react with the non-protonated hydroquinone, leading to the electro-polymerization of polymer chains. The electro-polymerization provides additional oxidation capacity, but it doesn't affect the reversibility of redox-active groups. The parasitic reaction of electro-polymerization can be suppressed by increasing the charge rate, as indicated by the Tafel plots (Supplementary Figure 6).

We have added the related discussion and figures in the revised Manuscript and Supplementary data, as below:

“As shown in Supplementary Figure 4, the CV curve of PHQ particulate suspension at the 50th cycle was nearly overlapped with that at the 1st and 2nd cycles, while the ratio of oxidation capacity to reduction capacity (Q_1/Q_2) were changed at different scan rates. The Q_1/Q_2 is 1.08 at 0.1 V/s and 1.34 at 0.025 V/s, respectively. The small oxidation peak at 1.2 V (vs. SHE) is proposed to be originated from the electro-polymerization of PHQ^{55,56}. As illustrated in Supplementary Figure 5, when the terminal hydroquinone was oxidized to the protonated benzoquinone in the electro-oxidation process, it might react with the non-protonated hydroquinone, leading to the electro-polymerization of polymer chains. The electro-polymerization can provide additional oxidation capacity, but it doesn't affect the reversibility of redox-active groups. The parasitic reaction of electro-polymerization can be suppressed by increasing the current rate, as indicated by the Tafel plots (Supplementary Figure 6).”

“The increasing Coulombic efficiency with increasing current densities may be attributed by the aforementioned suppression of the parasitic electro-polymerization reaction of PHQ at high current densities (Supplementary Figure 5 & 6).”

55. Yamamoto, K., Asada, T., Nishide, H. & Tsuchida, E. The preparation of poly (dihydroxyphenylene) through the electro-oxidative polymerization of hydroquinone. *B. Chem. Soc. Jpn.* **63**, 1211-1216 (1990).

56. Wang, P., Martin, B. D., Parida, S., Rethwisch, D. G. & Dordick, J. S. Multienzymic synthesis of poly(hydroquinone) for use as a redox polymer. *J. Am. Chem. Soc.* **117**, 12885-12886 (1995).

Supplementary Figure 4 | CV curves of 0.005 mol/ L PHQ particulates in 2.0 mol/L H₂SO₄ aqueous solution during the 1st, 2nd and 50th cycles at different scan rate of (a) 0.1 V/s and (b) 0.025 V/s, within a potential range from 0 to 1.5 V vs. SHE.

Supplementary Figure 5 | Proposed electro-polymerization process of PHQ in acidic solution.

Supplementary Figure 6 | Tafel plots of proposed parallel reactions of electro-polymerization and electro-oxidation of PHQ.

REVIEWERS' COMMENTS:

Reviewer #1 (Remarks to the Author):

Second Review for "All-Polymer Particulate Slurry Batteries"

The manuscript has been significantly improved for clarity of the introduced concept. The work advanced an interesting concept for redox battery by designing anolyte and catholyte materials whereby the active polymer particles undergo redox reactions, rather than through active molecules on inactive polymer surfaces. The approach is appealing to a broad audience, and the results are considered significant. After carefully and satisfactorily addressing comments made during the first review, this work is hereby deemed suitable for publication in this journal without further revision. The novelty has been clarified and several systematic experiments and explanations were added to expatiate on the technical concepts and the results.

Reviewer #2 (Remarks to the Author):

I can see that the authors took their efforts in improving their manuscript and answering the reviewers' questions. Basically, I can support the publication of this manuscript. However, I don't think I conveyed the meaning of my first question well enough. So I would like kindly ask the authors to address the following concerns in the final manuscript if it is to be published:

1. I was not asking what is the charge transferring mechanism within the polymer chain. I was curious how the electrons move between the interfaces of current collectors and the active materials. In my opinion, the electrons can only be transported to the outer circuit if the active materials are connected to the current collector in some ways. So that's also why I asked the functions of each component. My question is: is the electron transfer in the interface made possible by collision between the polymer particles and current collectors or simply because there are a lot of particles stuck in the porous structure of the carbon paper current collector? A post-run analysis of the cell would be good. E.g. a picture of the disassembled cell after the test.
2. For particle suspension flow batteries, the geometry of cell and the operating parameters such as pump rate can greatly affect the electrochemical performance of the cell. And the energy consumed by the peristaltic pump sometimes makes a great difference in the overall practicability of the system. How to reach the best balance is also a challenge. It is best the authors can discuss these aspects.

REVIEWERS' COMMENTS:

Reviewer #1 (Remarks to the Author):

Second Review for “All-Polymer Particulate Slurry Batteries”

The manuscript has been significantly improved for clarity of the introduced concept. The work advanced an interesting concept for redox battery by designing anolyte and catholyte materials whereby the active polymer particles undergo redox reactions, rather than through active molecules on inactive polymer surfaces. The approach is appealing to a broad audience, and the results are considered significant. After carefully and satisfactorily addressing comments made during the first review, this work is hereby deemed suitable for publication in this journal without further revision. The novelty has been clarified and several systematic experiments and explanations were added to expatiate on the technical concepts and the results.

Response: Thank you very much for reviewing our manuscript and also thanks for the very helpful suggestions on improving the manuscript.

Reviewer #2 (Remarks to the Author):

I can see that the authors took their efforts in improving their manuscript and answering the reviewers' questions. Basically, I can support the publication of this manuscript. However, I don't think I conveyed the meaning of my first question well enough. So I would like kindly ask the authors to address the following concerns in the final manuscript if it is to be published:

1. I was not asking what is the charge transferring mechanism within the polymer chain. I was curious how the electrons move between the interfaces of current collectors and the active materials. In my opinion, the electrons can only be transported to the outer circuit if the active materials are connected to the current collector in some ways. So that's also why I asked the functions of each component. My question is: is the electron transfer in the interface made possible by collision between the polymer particles and current collectors or simply because there are a lot of particles stuck in the porous structure of the carbon paper current collector? A post-run analysis of the cell would be good. E.g. a picture of the disassembled cell after the test.

Response: Thank you very much for your helpful question. We suggest the electrons can transport through the interface by collision between the polymer particulates and current collectors with the assistance of electric double layer (EDL) formed on the surface of particulates, as schematically illustrated in Fig. 1b. The photographs of disassembled PHQ/PI2 APPSB cell after long-term cycling is shown in Supplementary Fig. 14d, which shows only slight accumulation of polymer particulates on the carbon paper and in the flow channel. This indicates that the electron transfer is not relied on the particles stuck in the porous structure of the carbon paper current collector.

We have added the related discussion in the revised Manuscript and Supplementary Information:

“The electrons can transport through the interface by collision between the polymer particulates and

current collectors with the assistance of electric double layer (EDL) formed on the surface of particulates.”

“For PI2, the aggregation is worse than PI1 (Supplementary Fig. 14c). The photographs of disassembled PHQ/PI2 APPSB cell after long-term cycling is shown in Supplementary Fig. 14d, which shows slight accumulation of polymer particulates on the carbon paper and in the flow channel. The agglomerates precipitated in the flow channels and reservoirs may result in capacity fading after long-term tests. Therefore, to further improve the cycling stability of APPSBs, we suggest to introduce appropriate dispersion stabilizer in the particulate suspensions without the compromise of electrochemical performances, which will be an important aspect of our future research.”

Supplementary Figure 14. Post-run analysis of polymer particulate agglomeration degrees after long-term cycling. SEM images of (a) PHQ, (b) PI1, and (c) PI2 particulates after charging/discharging cycling. Scale bars, (a, b) 20 μm ; (c) 5 μm . (d) Optical photographs of disassembled PHQ/PI2 APPSB cell after long-term cycling test.

2. For particle suspension flow batteries, the geometry of cell and the operating parameters such as pump rate can greatly affect the electrochemical performance of the cell. And the energy consumed by the peristaltic pump sometimes makes a great difference in the overall practicability of the system. How to reach the best balance is also a challenge. It is best the authors can discuss these aspects.

Response: Thank you very much for your good question. Multiple dynamic models based on the fluid mechanics and electrochemistry have been proposed to achieve both optimal flow strategy and flow rate in all-vanadium RFBs^{R1,R2}. For large-scale system, the enhanced convective mass transport at higher pump rate can reduce the overpotential and increase the rate performance and energy density, yet it is at the cost of more pump power consumption. With the increase of the cell geometry and cell stacks, the practicability of the RFB system will be improved owing to higher device integrity and volume utilization. For APPSBs, in-depth studies on the rheological properties of concentrated particulate suspensions will be investigated in our future work, and the system efficiency of large-scale batteries shall be optimized to reach the best balance. We have added the related discussion in the

revised Manuscript:

“Especially, in-depth studies on the rheological properties of concentrated particulate suspensions will be investigated, and the system efficiency of large-scale batteries shall be optimized to reach the best balance.”

- [R1] Houser, J., Clement, J., Pezeshki, A. & Mench, M. M. Influence of architecture and material properties on vanadium redox flow battery performance. *J. Power Sources* **302**, 369–377 (2016).
- [R2] Xiao, W. & Tan, L. Control strategy optimization of electrolyte flow rate for all vanadium redox flow battery with consideration of pump. *Renew. Energ.* **133**, 1445–1454 (2019).